# Almost Optimal Model-Free Reinforcement Learning via Reference-Advantage Decomposition

**Zihan Zhang**
Department of Automation
Tsinghua University
zihan-zh17@mails.tsinghua.edu.cn

**Yuan Zhou**
Department of ISE
University of Illinois at Urbana-Champaign
yuanz@illinois.edu

**Xiangyang Ji**
Department of Automation
Tsinghua University
xyji@tsinghua.edu.cn

## Abstract

We study the reinforcement learning problem in the setting of finite-horizon episodic Markov Decision Processes (MDPs) with $S$ states, $A$ actions, and episode length $H$. We propose a model-free algorithm UCB-ADVANTAGE and prove that it achieves $\tilde{O}(\sqrt{H^2 SAT})$ regret where $T = KH$ and $K$ is the number of episodes to play. Our regret bound improves upon the results of [Jin et al., 2018] and matches the best known model-based algorithms as well as the information theoretic lower bound up to logarithmic factors. We also show that UCB-ADVANTAGE achieves low local switching cost and applies to concurrent reinforcement learning, improving upon the recent results of [Bai et al., 2019].

## 1 Introduction

Reinforcement learning (RL) [Burnetas and Katehakis, 1997] studies the problem where an agent aims to maximize its accumulative rewards through sequential decision making in an unknown environment modeled by Markov Decision Processes (MDPs). At each time step, the agent observes the current state $s$ and interacts with the environment by taking an action $a$ and transits to next state $s'$ following the underlying transition model.

There are mainly two types of algorithms to approach reinforcement learning: *model-based* and *model-free* learning. Model-based algorithms learn a model from the past experience and make decision based on this model while model-free algorithms only maintain a group of value functions and take the induced optimal actions. Because of these differences, model-free algorithms are usually more space- and time-efficient compared to model-based algorithms. Moreover, because of their simplicity and flexibility, model-free algorithms are popular in a wide range of practical tasks (e.g., DQN [Mnih et al., 2015], A3C [Mnih et al., 2016], TRPO [Schulman et al., 2015a], and PPO [Schulman et al., 2017]). On the other hand, however, it is believed that model-based algorithms may be able to take the advantage of the learned model and achieve better learning performance in terms of regret or sample complexity, which has been empirically evidenced by Deisenroth and Rasmussen [2011] and Schulman et al. [2015a]. Much experimental research has been done for both types of the algorithms, and given that there has been a long debate on their pros and cons that dates back to [Deisenroth and Rasmussen, 2011], a natural and intriguing theoretical question to study about reinforcement learning algorithms is that –

**Question 1.** *Is it possible that model-free algorithms achieve as competitive learning efficiency as model-based algorithms, while still maintaining low time and space complexities?*

Towards answering this question, the recent work by Jin et al. [2018] formally defines that an RL algorithm is model-free if its space complexity is always sublinear relative to the space required to store the MDP parameters, and then proposes a model-free algorithm (which is a variant of the $Q$-learning algorithm [Watkins, 1989]) that achieves the first $\sqrt{T}$-type regret bound for finite-horizon episodic MDPs in the tabular setting (i.e., discrete state spaces). However, there is still a gap of factor $\sqrt{H}$ between the regret of their algorithm and the best model-based algorithms. In this work, we close this gap by proposing a novel model-free algorithm, whose regret matches the optimal model-based algorithms, as well as the information theoretic lower bound. The results suggest that model-free algorithms can learn as efficiently as model-based ones, giving an affirmative answer to Question 1 in the setting of episodic tabular MDPs.

## 1.1 Our Results

**Main Theorem.** We propose a novel variant of the $Q$-learning algorithm, UCB-ADVANTAGE. We then prove the following main theorem of the paper.

**Theorem 1.** *For $T$ greater than some polynomial of $S$, $A$, and $H$, and for any $p \in (0,1)$, with probability $(1-p)$, the regret of UCB-ADVANTAGE is bounded by $\mathrm{Regret}(T) \leqslant \tilde{O}(\sqrt{H^2 SAT})$, where poly-logarithmic factors of $T$ and $1/p$ are hidden in the $\tilde{O}(\cdot)$ notation.*

Compared to the $\tilde{O}(\sqrt{H^3 SAT})$ regret bound of the UCB-Bernstein algorithm in [Jin et al., 2018], UCB-ADVANTAGE saves a factor of $\sqrt{H}$, and matches the information theoretic lower bound of $\Omega(\sqrt{H^2 SAT})$ in [Jin et al., 2018] up to logarithmic factors. The regret of UCB-ADVANTAGE is at the same order of the best model-based algorithms such as UCBVI [Azar et al., 2017] and vUCQ [Kakade et al., 2018].[1] However, the time complexity before time step $T$ is $O(T)$ and the space complexity is $O(SAH)$ for UCB-ADVANTAGE. In contrast, both UCBVI and vUCQ uses $\tilde{O}(TS^2A)$ time and $O(S^2AH)$ space.

One of the main technical ingredients of UCB-ADVANTAGE is to incorporate a novel update rule for the $Q$-function based on the proposed *reference-advantage decomposition*. More specifically, we propose to view the optimal value function $V^*$ as $V^* = V^{\mathrm{ref}} + (V^* - V^{\mathrm{ref}})$, where $V^{\mathrm{ref}}$, the *reference* component, is a comparably easier learned approximate of $V^*$ and the other component $(V^* - V^{\mathrm{ref}})$ is referred to as the *advantage* part. Based on this decomposition, the new update rule learns the corresponding parts of the $Q$-function using carefully designed (and different) subsets of the collected data, so as to minimize the deviation, maximize the data utilization, and reduce the estimation variance.

Another highlight of UCB-ADVANTAGE is the use of the *stage-based update framework* which enables an easy integration of the new update rule (as above) and the standard update rule. In such a framework, the visits to each state-action pair are partitioned into *stages*, which are used to design the trigger and subsets of data for each update.

**Implications.** An extra benefit of the stage-based update framework is to ensure the low frequency of policy switches of UCB-ADVANTAGE, stated as follows.

**Theorem 2.** *The local switching cost of UCB-ADVANTAGE is bounded by $O(SAH^2 \log T)$.*

While one may refer to Appendix C for the details of the theorem, the notion of local switching cost for RL is recently introduced and studied by Bai et al. [2019], where the authors integrate a lazy update scheme with the UCB-Bernstein algorithm [Jin et al., 2018] and achieve $\tilde{O}(\sqrt{H^3 SAT})$ regret and $O(SAH^3 \log T)$ local switching cost. In contrast, our result improves in both metrics of regret and switching cost.

Our results also apply to concurrent RL, a research direction closely related to batched learning and learning with low switching costs, stated as follows.

**Corollary 3.** *Given $M$ parallel machines, the concurrent and pure exploration version of UCB-ADVANTAGE can compute an $\epsilon$-optimal policy in $\tilde{O}(H^2 SA + H^3 SA/(\epsilon^2 M))$ concurrent episodes.*

In contrast, the state-of-the-art result [Bai et al., 2019] uses $\tilde{O}(H^3SA + H^4SA/(\epsilon^2 M))$ concurrent episodes. When $M = 1$, Corollary 3 implies that the single-threaded exploration version of UCB-ADVANTAGE uses $\tilde{O}(H^3SA/\epsilon^2)$ episodes to learn an $\epsilon$-optimal policy. In Appendix C, we provide a simple $\Omega(H^3SA/\epsilon^2)$-episode lower bound for the sample complexity, showing the optimality up to logarithmic factors.

## 1.2  Additional Related Works

**Regret Analysis for RL.**   Since our results focus on the tabular case, we will not mention most of the results on RL for continuous state spaces. For the tabular setting, there are plenty of recent works on model-based algorithms under various settings (e.g., [Jaksch et al., 2010, Agrawal and Jia, 2017, Azar et al., 2017, Ouyang et al., 2017, Fruit et al., 2019, Simchowitz and Jamieson, 2019, Zanette and Brunskill, 2019, Zhang and Ji, 2019]). The readers may refer to [Jin et al., 2018] for more detailed review and comparison. In contrast, fewer model-free algorithms are proposed. Besides [Jin et al., 2018], an earlier work [Strehl et al., 2006] implies that $T^{4/5}$-type regret can be achieved by a model-free algorithm.

**Variance Reduction and Advantage Functions.**   Variance reduction techniques via reference-advantage decomposition is used for faster optimization algorithms [Johnson and Zhang, 2013]. The technique is also recently applied to pure exploration in learning discounted MDPs [Sidford et al., 2018b,a]. However, since Sidford et al. [2018b,a] assume the access to a simulator and UCB-ADVANTAGE is completely online, our update rule and data partition design is very different. Our work is also the first for regret analysis in RL.

The use of advantage functions have also witnessed much success for RL in practice. For example, in A3C [Mnih et al., 2016], the advantage function is defined to be $\mathrm{Adv}(s, a) := Q^\pi(s, a) - V^\pi(s)$, and helps to reduce the estimation variance of the policy gradient. Similar definitions can also be found in other works such as [Sutton et al., 2000], Generalized Advantage Estimation [Schulman et al., 2015b] and Dueling DQN [Wang et al., 2015]. In comparison, our advantage function is defined on the states instead of the state-action pairs.

## 2  Preliminaries

We study the setting of episodic MDPs where an MDP is described by $(\mathcal{S}, \mathcal{A}, H, P, r)$. Here, $\mathcal{S} \times \mathcal{A}$ is the state-action space, $H$ is the length of each episode, $P$ is the transition probability matrix and $r$ is the deterministic reward function[2]. Without loss of generality, we assume that $r_h(s, a) \in [0, 1]$ for all $s, a, h$. During each episode, the agent observes the initial state $s_1$ which may be chosen by an *oblivious adversary* (i.e., the adversary may have the access to the algorithm description used by the agent but does not observe the execution trajectories of the agent[3]).

During each step within the episode, the agent takes an action $a_h$ and transits to $s_{h+1}$ according to $P_h(\cdot|s_h, a_h)$. The agent keeps running for $H$ steps and then the episode terminates.

A policy[4] $\pi$ is a mapping from $\mathcal{S} \times [H]$ to $\mathcal{A}$. Given a policy $\pi$, we define its value function and $Q$-function as

$$V_h^\pi(s) = \mathbb{E}\left[\sum_{h'=h}^{H} r_{h'}(s_{h'}, \pi_{h'}(s_{h'})) \Big| s_h = s, s_{h'+1} \sim P_{h'}(\cdot|s_{h'}, \pi_{h'}(s_{h'}))\right],$$

$$Q_h^\pi(s, a) = r_h(s, a) + P_h(\cdot|s, a)^\top V_{h+1}^\pi = r_h(s, a) + P_{s,a,h}V_{h+1}^\pi.$$

As boundary conditions, we define $V_{H+1}^\pi(s) = Q_{H+1}^\pi(s, a) = 0$ for any $\pi, s, a$. Also note that, for simplicity, throughout the paper, we use $xy$ to denote $x^T y$ for two vectors of the same dimension and use $P_{s,a,h}$ to denote $P_h(\cdot|s, a)$.

The optimal value function is then given by $V_h^*(s) = \sup_\pi V_h^\pi(s)$ and $Q_h^*(s,a) = r_h(s,a) + P_{s,a,h}V_{h+1}^*$ for any $(s,a) \in \mathcal{S} \times \mathcal{A}, h \in [H]$.

The learning problem consists of $K$ episodes, i.e, $T = KH$ steps. Let $s_1^k$ be the state given to the agent at the beginning of the $k$-th episode, and let $\pi_k$ be the policy adopted by the agent during the $k$-th episode. To goal is to minimize the total regret at time step $T$ which is defined as follows,

$$\text{Regret}(T) := \sum_{k=1}^K \left( V_1^*(s_1^k) - V_1^{\pi_k}(s_1^k) \right). \tag{1}$$

## 3  The UCB-ADVANTAGE Algorithm

In this section, we introduce the UCB-ADVANTAGE algorithm. We start by reviewing the $Q$-learning algorithms proposed in [Jin et al., 2018]. Recall that Jin et al. [2018] selects the learning rate $\alpha_t = \frac{H+1}{H+t}$, and sets the weights $\alpha_t^i = \alpha_i \Pi_{j=i+1}^t (1 - \alpha_j)$ for the $i$-th samples out of the a total of $t$ data points, for any state-action pair. Note that $\alpha_t^i$ is roughly $\Theta(H/t)$ for the indices $i \in \left[\frac{H-1}{H} \cdot t, t\right]$ and vanishes quickly when $i \ll \frac{H-1}{H} \cdot t$. As a result, their update process is roughly equivalent to using the latest $\frac{1}{H}$ fraction of samples to update the value function for any state-action pair. Next, we introduce our stage-based update framework, which shares much similarity with the process discussed above. However, our framework enjoys simpler analysis and enables easier integration of the two update rules which will be explained afterwards.

**Stages and Stage-Based Update Framework.**  For any triple $(s,a,h)$, we divide the samples received for the triple into consecutive *stages*. The length of each stage roughly increases exponentially with the growth rate $(1 + 1/H)$. More specifically, we define $e_1 = H$ and $e_{i+1} = \lfloor (1 + \frac{1}{H})e_i \rfloor$ for all $i \geqslant 1$, standing for the length of the stages. We also let $\mathcal{L} := \{\sum_{i=1}^j e_i | j = 1, 2, 3, \dots\}$ be the set of indices marking the ends of the stages.

Now we introduce the *stage-based update framework*. For any $(s,a,h)$ triple, we update $Q_h(s,a)$ when the total visit number of $(s,a,h)$ the end of the current stage (in other word, the total visit number occurs in $\mathcal{L}$). Only the samples in the latest stage will be used in this update. Using the language of [Jin et al., 2018], for any total visit number $t$ in the $(j+1)$-th stage, our update framework is equivalent to setting the weight distribution to be $\alpha_t^i = e_j^{-1} \cdot \mathbb{I}[i \text{ in the } j\text{-th stage}]$.

We note that the definition of stages is with respect to the triple $(s,a,h)$. For any fixed pair of $k$ and $h$, let $(s_h^k, a_h^k)$ be the state-action pair at the $h$-th step during the $k$-th episode of the algorithm. We say that $(k,h)$ falls in the $j$-th stage of $(s,a,h)$ if and only if $(s,a) = (s_h^k, a_h^k)$ and the total visit number of $(s_h^k, a_h^k)$ after the $k$-th episode is in $(\sum_{i=1}^{j-1} e_i, \sum_{i=1}^j e_i]$.

One benefit of our stage-based update framework is that it helps to reduce the number of the updates to the $Q$-function, leading to less local switching costs, which is recently also studied by Bai et al. [2019], where the authors propose to apply a lazy update scheme to the algorithms in Jin et al. [2018]. The lazy update scheme uses an exponential triggering sequence with a growth rate of $(1 + 1/(2H(H+1)))$, which is more conservative than the growth rate of stage lengths in our work. As a result, our algorithm saves an $H$ factor in the switching cost compared to [Bai et al., 2019].

More importantly, our stage-based update framework, compared to the algorithms in [Jin et al., 2018], (in our opinion) simplifies the analysis, makes it easier to integrate the standard update rule and the one based on the reference-advantage decomposition. Both update rules are used in our algorithm, and we now discuss them separately.

**The Standard Update Rule and its Limitation.**  The algorithms in [Jin et al., 2018] uses the following standard update rule,

$$Q_h(s,a) \leftarrow \widehat{P_{s,a,h}V_{h+1}} + r_h(s,a) + \overline{b}, \tag{2}$$

where $\overline{b}$ is the exploration bonus, and $\widehat{P_{s,a,h}V_{h+1}}$ is the empirical estimate of $P_{s,a,h}V_{h+1}$. We also adopt this update rule in our algorithm. However, a crucial restriction is that the earlier samples collected, the more deviation one would expect between the $V_{h+1}$ learned at that moment and the

true value. To ensure that these deviations do not ruin the whole estimate, we have to require that $\widehat{P_{s,a,h}V_{h+1}}$ only uses the samples acquired from the last stage. This means that we can only estimate the $P_{s,a,h}V_{h+1}$ term using about $1/H$ fraction of the obtained data, and we note that this is also the reason of the extra $\sqrt{H}$ occurred in the UCB-Bernstein algorithm by Jin et al. [2018].

**Reference-Advantage Decomposition and the Advantage-Based Update Rule.** We now introduce the reference-advantage decomposition, which is the key to reducing the extra $\sqrt{H}$ factor. At a high level, we aim at first learning a quite accurate estimation of the optimal value function $V^*$ and denote it by the *reference value function* $V^{\mathrm{ref}}$. The accuracy is controlled by an error parameter $\beta$ which is quite small but independent of $T$ or $K$. In other words, we wish to have $V_h^*(s) \leqslant V_h^{\mathrm{ref}}(s) \leqslant V_h^*(s) + \beta$ for all $s$ and $h$, and for the purpose of simple explanation, we set $\beta = 1/H$ at this moment; in our algorithm, $\beta$ can be any value that is less than $\sqrt{1/H}$ while independent of $T$ or $K$.

For starters, let us first assume that we have the access to the dreamed $V^{\mathrm{ref}}$ reference function as stated above. Now we write $V^* = V^{\mathrm{ref}} + (V^* - V^{\mathrm{ref}})$, and refer to the second term as the *advantage* compared to the reference values[5]. Now the $Q$-function can be updated using the following advantage-based rule,

$$Q_h(s,a) \leftarrow \widehat{P_{s,a,h}V_{h+1}^{\mathrm{ref}}} + \widehat{P_{s,a,h}(V_{h+1} - V_{h+1}^{\mathrm{ref}})} + r_h(s,a) + b, \qquad (3)$$

where $b$ is the exploration bonus, and both $\widehat{P_{s,a,h}V_{h+1}^{\mathrm{ref}}}$ and $\widehat{P_{s,a,h}(V_{h+1} - V_{h+1}^{\mathrm{ref}})}$ are empirical estimates of $P_{s,a,h}V_{h+1}^{\mathrm{ref}}$ and $P_{s,a,h}(V_{h+1} - V_{h+1}^{\mathrm{ref}})$ (respectively) based on the observed samples. We still have to require that $\widehat{P_{s,a,h}(V_{h+1} - V_{h+1}^{\mathrm{ref}})}$ uses the samples only from the last stage so as to limit the deviation error due to $V_{h+1}$ in the earlier samples.

Fortunately, thanks to the reference-advantage decomposition, and since that $V$ is learned based on $V^{\mathrm{ref}}$ and approximates $V^*$ even better than $V^{\mathrm{ref}}$, we have that $\|V_{h+1} - V_{h+1}^{\mathrm{ref}}\|_\infty \leqslant \beta = 1/H$ holds for all samples, which suffices to offset the weakness of using only $1/H$ of the total data, and helps to learn an accurate estimation of the second term. On the other hand, for the first term in the Right-Hand-Side of (3), since $V^{\mathrm{ref}}$ is fixed and never changes, we are able to use all the samples collected to conduct the estimation, without suffering any deviation. This means that the first term can also be estimated with high accuracy.

The discussion till now has assumed that the reference value vector $V^{\mathrm{ref}}$ is known. To remove this assumption, we note that $\beta$ is independent of $T$, therefore a natural hope is to learn $V^{\mathrm{ref}}$ using sample complexity also almost independent of $T$, incurring regret only in the lower order terms. However, since it is not always possible to learn the value function of every state (especially the ones almost not reachable), we need to integrate the learning for reference vector into the main algorithm, and much technical effort is made to enable the analysis for the integrated algorithm.

**Description of the Algorithm.** UCB-ADVANTAGE is described in Algorithm 1, where $c_1$, $c_2$, and $c_3$ are large enough positive universal constants so that concentration inequalities may be applied in the analysis. Besides the standard quantities such as $Q_h(s,a)$, $V_h(s)$, and the reference value function $V_h^{\mathrm{ref}}$, the algorithm keeps seven types of accumulators to facilitate the update to the $Q$- and value functions: accumulators $N_h(s,a)$ and $\check{N}_h(s,a)$ are used to keep the total visit number and the number of visits only counting the current stage to $(s,a,h)$, respectively. Three types of *intra-stage* accumulators are used for the samples in the latest stage; they are reset at the beginning of each stage and updated at every time step as follows (note that short-hands are defined for succinct presentation of the $Q$-function update rule in (9)):

$$\check{\mu} := \check{\mu}_h(s_h, a_h) \xleftarrow{+} V_{h+1}(s_{h+1}) - V_{h+1}^{\mathrm{ref}}(s_{h+1}); \qquad \check{v} := \check{v}_h(s_h, a_h) \xleftarrow{+} V_{h+1}(s_{h+1}); \qquad (4)$$

$$\check{\sigma} := \check{\sigma}_h(s_h, a_h) \xleftarrow{+} (V_{h+1}(s_{h+1}) - V_{h+1}^{\mathrm{ref}}(s_{h+1}))^2. \qquad (5)$$

Finally, the following two types of *global* accumulators are used for the samples in all stages,

$$\mu^{\mathrm{ref}} := \mu_h^{\mathrm{ref}}(s_h, a_h) \xleftarrow{+} V_{h+1}^{\mathrm{ref}}(s_{h+1}); \qquad \sigma^{\mathrm{ref}} := \sigma_h^{\mathrm{ref}}(s_h, a_h) \xleftarrow{+} (V_{h+1}^{\mathrm{ref}}(s_{h+1}))^2. \qquad (6)$$

**Algorithm 1** UCB-ADVANTAGE

---

**Initialize:** set all accumulators to 0; for all $(s, a, h) \in \mathcal{S} \times \mathcal{A} \times [H]$, set $V_h(s) \leftarrow H - h + 1$;
$Q_h(s, a) \leftarrow H - h + 1; V_h^{\mathrm{ref}}(s, a) \leftarrow H$;
**for** episodes $k \leftarrow 1, 2, \dots, K$ **do**
 observe $s_1$;
 **for** $h \leftarrow 1, 2, \dots, H$ **do**
  Take action $a_h \leftarrow \arg\max_a Q_h(s_h, a)$, and observe $s_{h+1}$.
  Update the accumulators via $n := N_h(s_h, a_h) \overset{+}{\leftarrow} 1, \check{n} := \check{N}_h(s_h, a_h) \overset{+}{\leftarrow} 1$, and (4), (5), (6).
  **if** $n \in \mathcal{L}$ {*Reaching the end of the stage and update triggered*} **then**
   {*Set the exploration bonuses, update the Q-function and the value function*}

   $$b \leftarrow c_1 \sqrt{\frac{\sigma^{\mathrm{ref}}/n - (\mu^{\mathrm{ref}}/n)^2}{n}\iota} + c_2 \sqrt{\frac{\check{\sigma}/\check{n} - (\check{\mu}/\check{n})^2}{\check{n}}\iota} + c_3 \big(\frac{H\iota}{n} + \frac{H\iota}{\check{n}} + \frac{H\iota^{\frac{3}{4}}}{n^{\frac{3}{4}}} + \frac{H\iota^{\frac{3}{4}}}{\check{n}^{\frac{3}{4}}}\big); \quad (7)$$

   $$\bar{b} \leftarrow 2\sqrt{\frac{H^2}{\check{n}}\iota}; \quad\quad (8)$$

   $$Q_h(s_h, a_h) \leftarrow \min\{r_h(s_h, a_h) + \frac{\check{v}}{\check{n}} + \bar{b},\ r_h(s_h, a_h) + \frac{\mu^{\mathrm{ref}}}{n} + \frac{\check{\mu}}{\check{n}} + b,\ Q_h(s_h, a_h)\}; \quad (9)$$

   $$V_h(s_h) \leftarrow \max_a Q_h(s_h, a); \quad\quad (10)$$

   $\check{N}_h(s_h, a_h), \check{\mu}_h(s_h, a_h), \check{v}_h(s_h, a_h), \check{\sigma}_h(s_h, a_h) \leftarrow 0$; {*Reset intra-stage accumulators*}
  **end if**
  **if** $\sum_a N_h(s_h, a) = N_0$ **then** $V_h^{\mathrm{ref}}(s_h) \leftarrow V_h(s_h)$; {*Learn the reference value function*}
 **end for**
**end for**

---

All accumulators are initialized to 0 at the beginning of the algorithm.

The algorithm sets $\iota \leftarrow \log(\frac{2}{p})$ (where $p$ is the parameter for the failure probability) and $\beta \leftarrow \frac{1}{\sqrt{H}}$. We also set $N_0 := \frac{c_4 S A H^5 \iota}{\beta^2}$ for a large enough universal constant $c_4 > 0$, denoting the number of visits needed for each state to learn a $\beta$-accurate reference value.

By the definition of the accumulators, the first two expressions in $\min\{\cdot\}$ in (9) respectively correspond to update rules (2) and (3), where $b$ and $\bar{b}$ are the respective exploration bonuses. The bonuses are set in a way that both expressions can be shown to upper bound $Q^*$ in the desired event. The update (9) also makes sure that the learned $Q$-function is non-increasing as the algorithm proceeds.

## 4 The Analysis (Proof of Theorem 1)

Let $N_h^k(s, a)$, $\check{N}_h^k(s, a)$, $Q_h^k(s, a)$, $V_h^k(s)$ and $V_h^{\mathrm{ref}, k}(s)$ respectively denote the values of $N_h(s, a)$, $\check{N}_h(s, a)$, $Q_h(s, a)$, $V_h(s)$ and $V_h^{\mathrm{ref}}(s)$ at the beginning of $k$-th episode. In particular, $N_h^{K+1}(s, a)$ denotes the number of visits of $(s, a, h)$ after all $K$ episodes are done.

To facilitate the proof, we need a few more notations. For each $k$ and $h$, let $n_h^k$ be the total number of visits to $(s_h^k, a_h^k, h)$ prior to the current stage with respect to the same triple. Let $\check{n}_h^k$ be the number of visits to $(s_h^k, a_h^k, h)$ during the stage immediately before the current stage. We let $l_{h,i}^k$ denote the index of the $i$-th episode among the $n_h^k$ episodes defined above. Also let $\check{l}_{h,i}^k$ be the index of the $i$-th episode among the $\check{n}_h^k$ episodes defined above. When $h$ and $k$ are clear from the context, we omit the two letters and use $l_i$ and $\check{l}_i$ for short. We use $\mu_h^{\mathrm{ref}, k}, \check{\mu}_h^k, \check{v}_h^k, \sigma_h^{\mathrm{ref}, k}, \check{\sigma}_h^k, b_h^k$ and $\bar{b}_h^k$ to denote respectively the values of $\mu^{\mathrm{ref}}, \check{\mu}, \check{v}, \sigma^{\mathrm{ref}}, \check{\sigma}, b$ and $\bar{b}$ in the computation of $Q_h^k(s_h^k, a_h^k)$ in (9).

Recall that the value function $Q_h(s, a)$ is non-increasing as the algorithm proceeds. On the other hand, we claim in the following proposition that $Q_h(s, a)$ upper bounds $Q_h^*(s, a)$ with high probability.

**Proposition 4.** *Let* $p \in (0, 1)$. *With probability at least* $(1 - 4T(H^2T^3 + 3))p$, *it holds that* $Q_h^*(s, a) \leqslant Q_h^{k+1}(s, a) \leqslant Q_h^k(s, a)$ *for any* $s, a, h, k$.

The proof of Proposition 4 involves some careful application of the concentration inequalities for martingales and is deferred to Appendix B.

## 4.1 Learning the Reference Value Function

As mentioned before, we hope to get an accurate estimate of $V^*$ as the reference value function. Similar to the proof of Lemma 2 in [Dong et al., 2019], we show in the following lemma (the proof of which deferred to Appendix B) that we can learn a good reference value for each state with bounded sample complexity. Also note that while it is possible to improve the upper bound in Lemma 5 via more refined analysis, the current form is sufficient to prove our main theorem.

**Lemma 5.** *Conditioned on the successful events of Proposition 4, for any $\epsilon \in (0, H]$, with probability $(1 - Tp)$ it holds that for any $h \in [H]$, $\sum_{k=1}^{K} \mathbb{I}\left[V_h^k(s_h^k) - V_h^*(s_h^k) \geqslant \epsilon\right] \leqslant O(SAH^5\iota/\epsilon^2)$.*

By Lemma 5 with $\epsilon$ set to $\beta$, the fact that $V^k$ is non-increasing in $k$ and the definition of $N_0$, we have the following corollary.

**Corollary 6.** *Conditioned on the successful events of Proposition 4 and Lemma 5, for every state $s$ we have that $n_h^k(s) \geqslant N_0 \implies V_h^*(s) \leqslant V_h^{\mathrm{ref},k}(s) \leqslant V_h^*(s) + \beta$.*

## 4.2 Regret Analysis with Reference-Advantage Decomposition

We now prove Theorem 1. We start by replacing $p$ by $p/\mathrm{poly}(H, T)$ so that we only need to show the desired regret bound with probability $(1 - \mathrm{poly}(H, T) \cdot p)$. The proof in this subsection will also be conditioned on the successful events in Proposition 4 and Lemma 5, so that the regret can be expressed as

$$\mathrm{Regret}(T) = \sum_{k=1}^{K} \left(V_1^*(s_1^k) - V_1^{\pi_k}(s_1^k)\right) \leqslant \sum_{k=1}^{K} \left(V_1^k(s_1^k) - V_1^{\pi_k}(s_1^k)\right). \tag{11}$$

Define $\delta_h^k := V_h^k(s_h^k) - V_h^*(s_h^k)$ and $\zeta_h^k := V_h^k(s_h^k) - V_h^{\pi_k}(s_h^k)$. Note that when $N_h^k(s_h^k, a_h^k) \in \mathcal{L}$, we have that $n_h^k = N_h^k(s_h^k, a_h^k)$ and $\check{n}_h^k = \check{N}_h^k(s_h^k, a_h^k)$. Following the update rules (9) and (10), we have that[6]

$$V_h^k(s_h^k) \leqslant \mathbb{I}\left[n_h^k = 0\right] H + r_h(s_h^k, a_h^k) + \frac{\mu_h^{\mathrm{ref},k}}{n_h^k} + \frac{\check{\mu}_h^k}{\check{n}_h^k} + b_h^k$$

$$= \mathbb{I}\left[n_h^k = 0\right] H + r_h(s_h^k, a_h^k) + \frac{1}{n_h^k}\sum_{i=1}^{n_h^k} V_{h+1}^{\mathrm{ref},l_i}(s_{h+1}^{l_i}) + \frac{1}{\check{n}_h^k}\sum_{i=1}^{\check{n}_h^k}\left(V_{h+1}^{\check{l}_i}(s_{\check{l}_i,h+1}) - V_{h+1}^{\mathrm{ref},\check{l}_i}(s_{\check{l}_i,h+1})\right) + b_h^k.$$

Together with the Bellman equation $V_h^{\pi_k}(s_h^k) = r_h(s_h^k, a_h^k) + P_{s_h^k, a_h^k, h}V_{h+1}^{\pi_k}$, we have that

$$\zeta_h^k = V_h^k(s_h^k) - V_h^{\pi_k}(s_h^k)$$

$$\leqslant \mathbb{I}\left[n_h^k = 0\right] H + \frac{1}{n_h^k}\sum_{i=1}^{n_h^k} V_{h+1}^{\mathrm{ref},l_i}(s_{h+1}^{l_i}) + \frac{1}{\check{n}_h^k}\sum_{i=1}^{\check{n}_h^k}\left(V_{h+1}^{\check{l}_i}(s_{\check{l}_i,h+1}) - V_{h+1}^{\mathrm{ref},\check{l}_i}(s_{\check{l}_i,h+1})\right)$$

$$\qquad + b_h^k - P_{s_h^k, a_h^k, h}V_{h+1}^{\pi_k}$$

$$\leqslant \mathbb{I}\left[n_h^k = 0\right] H + \frac{1}{n_h^k}\sum_{i=1}^{n_h^k} P_{s_h^k, a_h^k, h}V_{h+1}^{\mathrm{ref},l_i} + \frac{1}{\check{n}_h^k}\sum_{i=1}^{\check{n}_h^k} P_{s_h^k, a_h^k, h}(V_{h+1}^{\check{l}_i} - V_{h+1}^{\mathrm{ref},\check{l}_i})$$

$$\qquad + 2b_h^k - P_{s_h^k, a_h^k, h}V_{h+1}^{\pi_k} \tag{12}$$

$$= \mathbb{I}\left[n_h^k = 0\right] H + P_{s_h^k, a_h^k, h}\Big(\frac{1}{n_h^k}\sum_{i=1}^{n_h^k} V_{h+1}^{\mathrm{ref},l_i} - \frac{1}{\check{n}_h^k}\sum_{i=1}^{\check{n}_h^k} V_{h+1}^{\mathrm{ref},\check{l}_i}\Big)$$

$$+ P_{s_h^k, a_h^k, h}\big(\frac{1}{\check{n}_h^k} \sum_{i=1}^{\check{n}_h^k} (V_{h+1}^{\check{l}_i} - V_{h+1}^*)\big) + P_{s_h^k, a_h^k, h}(V_{h+1}^* - V_{h+1}^{\pi_k}) + 2b_h^k$$

$$\leqslant \mathbb{I}\big[n_h^k = 0\big] H + \frac{1}{\check{n}_h^k} \sum_{i=1}^{\check{n}_h^k} \delta_{h+1}^{\check{l}_i} - \delta_{h+1}^k + \zeta_{h+1}^k + \underbrace{\psi_{h+1}^k + \xi_{h+1}^k + \phi_{h+1}^k + 2b_h^k}_{\Lambda_{h+1}^k}, \tag{13}$$

where letting $V^{\mathrm{REF}}$ be the final reference vector (i.e., $V^{\mathrm{REF}} := V^{\mathrm{ref}, K+1}$), and $\mathbf{1}_j$ be the $j$-th canonical basis vector (i.e., $(0, \dots, 0, 1, 0, \dots, 0)$ where the only 1 is located at the $j$-th entry), we define

$$\psi_{h+1}^k := \frac{1}{n_h^k} \sum_{i=1}^{n_h^k} P_{s_h^k, a_h^k, h}(V_{h+1}^{\mathrm{ref}, l_i} - V_{h+1}^{\mathrm{REF}}), \quad \xi_{h+1}^k := \frac{1}{\check{n}_h^k} \sum_{i=1}^{\check{n}_h^k} (P_{s_h^k, a_h^k, h} - \mathbf{1}_{s_{h+1}^{\check{l}_i}})(V_{h+1}^{\check{l}_i} - V_{h+1}^*),$$

$$\phi_{h+1}^k := (P_{s_h^k, a_h^k, h} - \mathbf{1}_{s_{h+1}^k})(V_{h+1}^* - V_{h+1}^{\pi_k}).$$

Here at Inequality (12) is implied by the successful event of martingale concentration (which is implied by the successful event in the proof of Proposition 4, in particular, Inequality (45)). Inequality (13) holds by the fact that $V_{h+1}^{\mathrm{ref}, k} \geqslant V_{h+1}^{\mathrm{REF}}$ for any $k, h$. Now we turn to bound $\sum_{k=1}^K \zeta_h^k$. Note that

$$\sum_{k=1}^K \zeta_h^k \leqslant \sum_{k=1}^K \mathbb{I}\big[n_h^k = 0\big] H + \sum_{k=1}^K \big(\frac{1}{\check{n}_h^k} \sum_{i=1}^{\check{n}_h^k} \delta_{h+1}^{\check{l}_{h,i}}\big) + \sum_{k=1}^K (\zeta_{h+1}^k + \Lambda_{h+1}^k - \delta_{h+1}^k). \tag{14}$$

The first term in the **RHS** of (14) is bounded by $\sum_{k=1}^K \mathbb{I}[n_h^k = 0] \leqslant SAH$ because $n_h^k \geqslant H$ when $N_h^k(s_h^k, a_h^k) \geqslant H$. We rewrite the second term as

$$\sum_{k=1}^K \big(\frac{1}{\check{n}_h^k} \sum_{i=1}^{\check{n}_h^k} \delta_{h+1}^{\check{l}_{h,i}}\big) = \sum_{k=1}^K \frac{1}{\check{n}_h^k} \sum_{j=1}^K \delta_{h+1}^j \sum_{i=1}^{\check{n}_h^k} \mathbb{I}[j = \check{l}_{h,i}^k] = \sum_{j=1}^K \delta_{h+1}^j \sum_{k=1}^K \frac{1}{\check{n}_h^k} \sum_{i=1}^{\check{n}_h^k} \mathbb{I}[j = \check{l}_{h,i}^k]. \tag{15}$$

Let $j \geqslant 1$ be a fixed episode. Note that $\sum_{i=1}^{\check{n}_h^k} \mathbb{I}[j = \check{l}_{h,i}^k] = 1$ if and only if $(s_h^j, a_h^j) = (s_h^k, a_h^k)$, and $(j, h)$ falls in the previous stage that $(k, h)$ falls in. As a result, every $k$ such that $\sum_{i=1}^{\check{n}_h^k} \mathbb{I}[j = \check{l}_{h,i}^k] = 1$ has the same $\check{n}_h^k$ which we denote by $Z_j$, and the set $\{k : \sum_{i=1}^{\check{n}_h^k} \mathbb{I}[j = \check{l}_{h,i}^k] = 1\}$ has at most $(1 + \frac{1}{H})Z_j$ elements. Therefore, for every $j$ we have that

$$\sum_{k=1}^K \frac{1}{\check{n}_h^k} \sum_{i=1}^{\check{n}_h^k} \mathbb{I}[j = \check{l}_{h,i}^k] \leqslant 1 + \frac{1}{H}. \tag{16}$$

Because $\delta_{h+1}^k \leqslant \zeta_{h+1}^k$, combining (14), (15), and (16), we have that

$$\sum_{k=1}^K \zeta_h^k \leqslant SAH^2 + (1 + \frac{1}{H}) \sum_{k=1}^K \delta_{h+1}^k - \sum_{k=1}^K \delta_{h+1}^k + \sum_{k=1}^K \zeta_{h+1}^k + \sum_{k=1}^K \Lambda_{h+1}^k$$

$$\leqslant SAH^2 + (1 + \frac{1}{H}) \sum_{k=1}^K \zeta_{h+1}^k + \sum_{k=1}^K \Lambda_{h+1}^k. \tag{17}$$

Iterating the derivation above for $h = 1, 2, \cdots, H$ and we have that

$$\sum_{k=1}^K \zeta_1^k \leqslant O\Big(SAH^3 + \sum_{h=1}^H \sum_{k=1}^K (1 + \frac{1}{H})^{h-1} \Lambda_{h+1}^k\Big). \tag{18}$$

We bound $\sum_{h=1}^H \sum_{k=1}^K (1 + \frac{1}{H})^{h-1} \Lambda_{h+1}^k$ in the lemma below. The detailed proof is deferred to Appendix B due to space constraints.

**Lemma 7.** *With probability at least $(1 - O(H^2 T^4 p))$, it holds that*

$$\sum_{h=1}^{H} \sum_{k=1}^{K} (1 + \frac{1}{H})^{h-1} \Lambda_{h+1}^k = O\left(\sqrt{SAH^2 T\iota} + H\sqrt{T\iota}\log(T) + S^2 A^{\frac{3}{2}} H^8 \iota T^{\frac{1}{4}}\right). \quad (19)$$

Combining Proposition 4, Lemma 5, (18) and Lemma 7, we conclude that with probability at least $(1 - O(H^2 T^4 p))$,

$$\text{Regret}(T) = \sum_{k=1}^{K} \zeta_1^k = O\left(\sqrt{SAH^2 T\iota} + H\sqrt{T\iota}\log(T) + S^2 A^{\frac{3}{2}} H^8 \iota T^{\frac{1}{4}}\right).$$

## Broader Impact

This work is theoretical and a broader impact discussion is not applicable.

## Acknowledgements

Zihan Zhang and Xiangyang Ji are supported by the National Key R&D Program of China under Grant 2018AAA0102800 and 2018AAA0102801. Yuan Zhou is supported in part by NSF CCF-2006526, a Ye Grant and a JPMorgan Chase AI Research Faculty Research Award.

## Footnotes

[1] Both Azar et al. [2017] and Kakade et al. [2018] assume equal transition matrices $P_1 = P_2 = \cdots = P_H$. In this work, we adopt the same setting as in, e.g., [Jin et al., 2018] and [Bai et al., 2019], where $P_1, P_2, \ldots, P_H$ can be different. This adds a factor of $\sqrt{H}$ to the regret analysis in [Azar et al., 2017] and [Kakade et al., 2018].

[2]Our results generalize to stochastic reward functions easily.

[3]Another adversary model is the the stronger *adaptive adversary* who may observe the execution trajectories and select the initial states based on the observation. While it is possible that a more careful analysis of our algorithm also works for the adaptive adversary, we do not make any effort verifying this statement. We also note that previous works such as [Jin et al., 2018, Bai et al., 2019] do not explicitly define their adversary models and it is not clear whether their analysis works for the adaptive adversary.

[4]In this work, we mainly consider deterministic policies since the optimal value function can be achieved by a deterministic policy.

[5]Interestingly, one might argue that the term should rather be called "disadvantage" as it is always non-positive. We choose the name "advantage" to highlight the similarity between our algorithm and many empirical algorithms in literature. See Section 1.2 for more discussion.

[6]Here we define $0/0$ to be $0$ so that forms such as $\frac{1}{n_h^k}\sum_{i=1}^{n_h^k} X_i$ are treated as $0$ if $n_h^k = 0$.

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
