[Supplementary Material]

# Appendices

Table 1: Explanation of the notations

| | |
|---|---|
| $\{V_h^k\}(\{Q_h^k\})$ | the value ($Q$-) function at the beginning of the $k$-th episode; |
| $\{V_h^{\text{ref},k}\}$ | the reference value function at the beginning of the $k$-th episode; |
| $\{N_h^k(s,a)\}$ | the number of visits to $(s,a,h)$ before the beginning of the $k$-th episode; |
| $\{\check{N}_h^k(s,a)\}$ | the number of visits to $(s,a,h)$ in the current stage (at the beginning of the $k$-th episode) with respect to the same triple; |
| $\{\mu_h^{\text{ref},k}\}(\{\sigma_h^{\text{ref},k}\})$ | the accumulator for the mean (variance) of the reference value function; |
| $\{\check{\mu}_h^k\}(\{\check{\sigma}_h^k\})$ | the accumulator for the mean (variance) of the advantage (i.e., the difference between the value and the reference value function) in the current stage; |
| $\{\check{v}_h^k\}$ | the accumulator for the mean of the value function in the current stage; |
| $\{b_h^k\}, \{b_h^k\}$ | the exploration bonuses for the two types of updates; |
| $\{V_h^{\text{REF}}\}$ | the final reference value function; |
| $\{V_h^*\}(\{Q_h^*\})$ | the optimal value ($Q$-) function; |
| $n_h^k$ | the number of visits to $(s_h^k, a_h^k, h)$ before the current stage with respect to the same triple; |
| $\check{n}_h^k$ | the number of visits to $(s_h^k, a_h^k, h)$ in the stage immediately before the current stage with respect to the same triple; |
| $l_{h,i}^k$ | the index (time step) of the $i$-th episode among the $n_h^k$ episodes of that visited $(s_h^k, a_h^k, h)$ before the current stage with respect to the same triple; |
| $\check{l}_{h,i}^k$ | the index (time step) of the $i$-th episode among the $\check{n}_h^k$ episodes in the stage immediately before the current stage with respect to $(s_h^k, a_h^k, h)$. |

## A Basic Lemmas

**Lemma 8** (Azuma-Hoeffding Inequality). *Suppose $\{X_k\}_{k=0,1,2,...}$ is a martingale and $|X_k - X_{k-1}| \leqslant c_k$ almost surely. Then for all positive integers $N$ and all positive reals $\epsilon$, it holds that*

$$\mathbb{P}\left[|X_N - X_0| \geqslant \epsilon\right] \leqslant 2\exp\left(\frac{-\epsilon^2}{2\sum_{k=1}^N c_k^2}\right).$$

**Lemma 9** (Freedman's Inequality, Theorem 1.6 of [Freedman et al., 1975]). *Let $(M_n)_{n\geqslant 0}$ be a martingale such that $M_0 = 0$ and $|M_n - M_{n-1}| \leqslant c$. Let $\text{Var}_n = \sum_{k=1}^n \mathbb{E}[(M_k - M_{k-1})^2|\mathcal{F}_{k-1}]$ for $n \geqslant 0$, where $\mathcal{F}_k = \sigma(M_0, M_1, M_2, ..., M_k)$. Then, for any positive $x$ and for any positive $y$,*

$$\mathbb{P}\left[\exists n : M_n \geqslant x \text{ and } \text{Var}_n \leqslant y\right] \leqslant \exp\left(-\frac{x^2}{2(y+cx)}\right). \tag{20}$$

**Lemma 10.** *Let $(M_n)_{n\geqslant 0}$ be a martingale such that $M_0 = 0$ and $|M_n - M_{n-1}| \leqslant c$ for some $c > 0$ and any $n \geqslant 1$. Let $\text{Var}_n = \sum_{k=1}^n \mathbb{E}[(M_k - M_{k-1})^2|\mathcal{F}_{k-1}]$ for $n \geqslant 0$, where $\mathcal{F}_k = \sigma(M_1, M_2, ..., M_k)$. Then for any positive integer $n$, and any $\epsilon, p > 0$, we have that*

$$\mathbb{P}\left[|M_n| \geqslant 2\sqrt{\text{Var}_n \log(\frac{1}{p})} + 2\sqrt{\epsilon\log(\frac{1}{p})} + 2c\log(\frac{1}{p})\right] \leqslant \left(\frac{2nc^2}{\epsilon} + 2\right)p. \tag{21}$$

*Proof.* For any fixed $n$, we apply Lemma 9 with $y = i\epsilon$ and $x = \pm(2\sqrt{y\log(\frac{1}{p})} + 2c\log(\frac{1}{p}))$. For each $i = 1, 2, \ldots, \lceil\frac{nc^2}{\epsilon}\rceil$, we get that

$$\mathbb{P}\left[|M_n| \geqslant 2\sqrt{(i-1)\epsilon\log(\frac{1}{p})} + 2\sqrt{\epsilon\log(\frac{1}{p})} + 2c\log(\frac{1}{p}), \text{Var}_n \leqslant i\epsilon\right]$$

$$\leqslant \mathbb{P}\left[|M_n| \geqslant 2\sqrt{i\epsilon\log(\frac{1}{p})} + 2c\log(\frac{1}{p}), \text{Var}_n \leqslant i\epsilon\right]$$

$$\leqslant 2p. \tag{22}$$

Then via a union bound, we have that

$$\mathbb{P}\left[|M_n| \geqslant 2\sqrt{\mathrm{Var}_n \log(\frac{1}{p})} + 2\sqrt{\epsilon \log(\frac{1}{p})} + 2c\log(\frac{1}{p})\right]$$

$$\leqslant \sum_{i=1}^{\lceil \frac{nc^2}{\epsilon} \rceil} \mathbb{P}\left[|M_n| \geqslant 2\sqrt{(i-1)\epsilon \log(\frac{1}{p})} + 2\sqrt{\epsilon \log(\frac{1}{p})} + 2c\log(\frac{1}{p}), (i-1)\epsilon \leqslant \mathrm{Var}_n \leqslant i\epsilon\right]$$

$$\leqslant \sum_{i=1}^{\lceil \frac{nc^2}{\epsilon} \rceil} \mathbb{P}\left[|M_n| \geqslant 2\sqrt{(i-1)\epsilon \log(\frac{1}{p})} + 2\sqrt{\epsilon \log(\frac{1}{p})} + 2c\log(\frac{1}{p}), \mathrm{Var}_n \leqslant i\epsilon\right]$$

$$\leqslant \left(\frac{2nc^2}{\epsilon} + 2\right)p. \tag{23}$$

$\square$

**Lemma 11.** *For any non-negative weights $\{w_h(s,a)\}_{s\in\mathcal{S},a\in\mathcal{A},h\in[H]}$ and $\alpha \in (0,1)$, it holds that*

$$\sum_{k=1}^{K}\sum_{h=1}^{H} \frac{w_h(s_h^k, a_h^k)}{(n_h^k)^\alpha} \leqslant \frac{2^\alpha}{1-\alpha}\sum_{s,a,h} w_h(s,a)(N_h^{K+1}(s,a))^{1-\alpha}, \tag{24}$$

*and*

$$\sum_{k=1}^{K}\sum_{h=1}^{H} \frac{w_h(s_h^k, a_h^k)}{(\check{n}_h^k)^\alpha} \leqslant \frac{2^{2\alpha}H^\alpha}{1-\alpha}\sum_{s,a,h} w_h(s,a)(N_h^{K+1}(s,a))^{1-\alpha}.$$

*In the case $\alpha = 1$, it holds that*

$$\sum_{k=1}^{K}\sum_{h=1}^{H} \frac{w_h(s_h^k, a_h^k)}{n_h^k} \leqslant 2\sum_{s,a,h} w_h(s,a)\log(N_h^{K+1}(s,a)), \tag{25}$$

*and*

$$\sum_{k=1}^{K}\sum_{h=1}^{H} \frac{w_h(s_h^k, a_h^k)}{\check{n}_h^k} \leqslant 4H\sum_{s,a,h} w_h(s,a)\log(N_h^{K+1}(s,a)).$$

*Proof.* By the definition of $\mathcal{L}$, for any $h,k$ such that $n_h^k > 0$, there exists $j$ such that $\check{n}_h^k = e_j$ and $n_h^k = \sum_{i=1}^{j} e_i$. Therefore, $\frac{1}{2H}n_h^k \leqslant \check{n}_h^k \leqslant \frac{3}{H}n_h^k$. So it suffices to prove (24) and (25). By basic calculus, for two positive numbers $x,y$ such that $y/2 \leqslant x \leqslant y$ and any $\alpha \in (0,1)$, we have that

$$y^{1-\alpha} - x^{1-\alpha} \geqslant (1-\alpha)(y-x)y^{-\alpha} \geqslant (1-\alpha)(y-x)2^{-\alpha}x^{-\alpha}, \tag{26}$$

and

$$\log(y) - \log(x) \geqslant \frac{y-x}{y} \geqslant 2\frac{y-x}{x}. \tag{27}$$

By applying (26) and (27) with $y = \sum_{i=1}^{j+1} e_i$ and $x = \sum_{i=1}^{j} e_i$ for $j = 1, 2, ...$ and taking sum, we have

$$\sum_{k=1}^{K}\sum_{h=1}^{H} \frac{w_h(s_h^k, a_h^k)}{(n_h^k)^\alpha} \leqslant \sum_{s,a,h} w_h(s,a) \sum_{j:\sum_{i=1}^{j} e_i \leqslant N_h^{K+1}(s,a)} \frac{\min\{e_{j+1}, N_h^{K+1}(s,a) - \sum_{i=1}^{j} e_i\}}{(\sum_{i=1}^{j} e_j)^\alpha}$$

$$\leqslant \frac{2^\alpha}{1-\alpha}\sum_{s,a,h} w_h(s,a)(N_h^{K+1}(s,a))^{1-\alpha}$$

and

$$\sum_{k=1}^{K}\sum_{h=1}^{H} \frac{w_h(s_h^k, a_h^k)}{n_h^k} \leqslant \sum_{s,a,h} w_h(s,a) \sum_{j:\sum_{i=1}^{j} e_i \leqslant N_h^{K+1}(s,a)} \frac{\min\{e_{j+1}, N_h^{K+1}(s,a) - \sum_{i=1}^{j} e_i\}}{(\sum_{i=1}^{j} e_j)}$$

$$\leqslant 2\sum_{s,a,h} w_h(s,a)\log(N_h^{K+1}(s,a)).$$

$\square$

# B  Missing Proofs in the Regret Analysis

## B.1  Proof of Proposition 4

We prove $Q_h^*(s,a) \leqslant Q_h^k(s,a)$ for all $k,h,s,a$, by induction on $k$. Firstly, the conclusion holds when $k=1$. For $k \geqslant 2$, assume $Q_h^*(s,a) \leqslant Q_h^u(s,a)$ for any $h,s,a$ and $1 \leqslant u \leqslant k$. Let $(s,a,h)$ be fixed. If we do not update $Q_h(s,a)$ in the $k$-th episode, then $Q_h^{k+1}(s,a) = Q_h^k(s,a) \geqslant Q_h^*(s,a)$. Otherwise, we have

$$Q_h^{k+1}(s,a) = \min\Bigg\{ \underbrace{r_h(s,a) + \frac{\underline{\mu}^{\mathrm{ref}}}{\underline{n}} + \frac{\check{\underline{\mu}}}{\check{\underline{n}}} + \underline{b}}_{(i)}, \quad \underbrace{r_h(s,a) + \frac{\check{\underline{v}}}{\check{\underline{n}}} + \overline{b}}_{(ii)}, \quad Q_h^k(s,a) \Bigg\}, \tag{28}$$

where $\underline{\mu}^{\mathrm{ref}}$, $\check{\underline{\mu}}$, $\underline{\sigma}^{\mathrm{ref}}$, $\check{\underline{\sigma}}$, $\underline{n}$, $\check{\underline{n}}$, $\underline{b}$ and $\overline{b}$ are given by respectively the values of $\mu^{\mathrm{ref}}$, $\check{\mu}$, $\sigma^{\mathrm{ref}}$, $\sigma$, $n$, $\check{n}$, $b$ and $\overline{b}$ to compute $Q_h^{k+1}(s,a)$ in (9). We use $\underline{l}_i$ to denote the episode index of the $i$-th sample and $\check{\underline{l}}_i$ to denote the episode index of the $i$-th sample of the last stage with respect to the triple $(s,a,h)$. Besides the last $Q_h^k(s,a)$ term, there are two non-trivial cases to discuss (corresponding to $(i)$ and $(ii)$).

*For the first case,* we have that

$$Q_h^{k+1}(s,a) = r_h(s,a) + \frac{\underline{\mu}^{\mathrm{ref}}}{\underline{n}} + \frac{\check{\underline{\mu}}}{\check{\underline{n}}} + \underline{b}$$

$$= r_h(s,a) + P_{s,a,h}\left(\frac{1}{\underline{n}}\sum_{i=1}^{\underline{n}} V_{h+1}^{\mathrm{ref},\underline{l}_i}\right) + P_{s,a,h}\left(\frac{1}{\check{\underline{n}}}\sum_{i=1}^{\check{\underline{n}}}(V_{h+1}^{\check{\underline{l}}_i} - V_{h+1}^{\mathrm{ref},\check{\underline{l}}_i})\right) + \chi_1 + \chi_2 + \underline{b}$$

$$\geqslant r_h(s,a) + P_{s,a,h}\left(\frac{1}{\check{\underline{n}}}\sum_{i=1}^{\check{\underline{n}}} V_{h+1}^{\check{\underline{l}}_i}\right) + \chi_1 + \chi_2 + \underline{b} \tag{29}$$

$$\geqslant r_h(s,a) + P_{s,a,h} V_{h+1}^* + \chi_1 + \chi_2 + \underline{b} \tag{30}$$

$$= Q_h^*(s,a) + \chi_1 + \chi_2 + \underline{b}$$

where

$$\chi_1 := \frac{1}{\underline{n}}\sum_{i=1}^{\underline{n}}\left(V_{h+1}^{\mathrm{ref},\underline{l}_i}(s_{h+1}^{\underline{l}_i}) - P_{s,a,h}V_{h+1}^{\mathrm{ref},\underline{l}_i}\right), \tag{31}$$

$$W_{h+1}^l := V_{h+1}^l - V_{h+1}^{\mathrm{ref},l}, \quad \forall l \geqslant 1 \tag{32}$$

$$\chi_2 := \frac{1}{\check{\underline{n}}}\sum_{i=1}^{\check{\underline{n}}}\left(W_{h+1}^{\check{\underline{l}}_i}(s_{h+1}^{\check{\underline{l}}_i}) - P_{s,a,h}W_{h+1}^{\check{\underline{l}}_i}\right). \tag{33}$$

Here, Inequality (29) holds because $V_{h+1}^{\mathrm{ref},u}$ is non-increasing in $u$, Inequality (30) is by the induction $V^u \geqslant V^*$ for any $1 \leqslant u \leqslant k$.

Define $\mathbb{V}(x,y) := x^\top(y^2) - (x^\top y)^2$ for two vectors $x,y$ of the same dimension, where $y^2$ is obtained by squaring each entry of $y$. By Lemma 10 with $\epsilon = \frac{1}{T^2}$, we have that with probability $(1 - 2(H^2T^3 + 1)p)$ it holds that

$$|\chi_1| \leqslant 2\sqrt{\frac{(\sum_{i=1}^{n}\mathbb{V}(P_{s,a,h},V_{h+1}^{\mathrm{ref},\underline{l}_i}))\iota}{\underline{n}^2}} + 2\frac{\sqrt{\iota}}{T\underline{n}} + \frac{2H\iota}{\underline{n}} \tag{34}$$

$$|\chi_2| \leqslant 2\sqrt{\frac{(\sum_{i=1}^{\check{n}}\mathbb{V}(P_{s,a,h},W_{h+1}^{\check{\underline{l}}_i}))\iota}{\check{\underline{n}}}} + 2\frac{\sqrt{\iota}}{T\check{\underline{n}}} + \frac{2H\iota}{\check{\underline{n}}}. \tag{35}$$

We now bound $\sum_{i=1}^{\check{n}}\mathbb{V}(P_{s,a,h},V_{h+1}^{\mathrm{ref},\underline{l}_i})$ in order to upper bound $|\chi_1|$. Define

$$\underline{\nu}^{\mathrm{ref}} := \frac{\underline{\sigma}^{\mathrm{ref}}}{\underline{n}} - \left(\frac{\underline{\mu}^{\mathrm{ref}}}{\underline{n}}\right)^2.$$

We claim that,

**Lemma 12.** *With probability* $(1-2p)$, *it holds that*

$$\sum_{i=1}^{\underline{n}} \mathbb{V}(P_{s,a,h}, V_{h+1}^{\mathrm{ref},l_i}) \leqslant \underline{n} \cdot \underline{\nu}^{\mathrm{ref}} + 3H^2\sqrt{\underline{n}\iota}. \tag{36}$$

*Proof.* We have that

$$\sum_{i=1}^{\underline{n}} \mathbb{V}(P_{s,a,h}, V_{h+1}^{\mathrm{ref},l_i}) := \sum_{i=1}^{\underline{n}} \left( P_{s,a,h}(V_{h+1}^{\mathrm{ref},l_i})^2 - (P_{s,a,h}V_{h+1}^{\mathrm{ref},l_i})^2 \right)$$

$$= \sum_{i=1}^{\underline{n}} (V_{h+1}^{\mathrm{ref},l_i}(s_{h+1}^{l_i}))^2 - \frac{1}{\underline{n}}\left( \sum_{i=1}^{n} V_{h+1}^{\mathrm{ref},l_i}(s_{h+1}^{l_i}) \right)^2 + \chi_3 + \chi_4 + \chi_5$$

$$= \underline{n} \cdot \underline{\nu}^{\mathrm{ref}} + \chi_3 + \chi_4 + \chi_5, \tag{37}$$

where

$$\chi_3 := \sum_{i=1}^{n} \left( P_{s,a,h}(V_{h+1}^{\mathrm{ref},l_i})^2 - (V_{h+1}^{\mathrm{ref},l_i}(s_{h+1}^{l_i}))^2 \right), \tag{38}$$

$$\chi_4 := \frac{1}{\underline{n}}\left( \sum_{i=1}^{\underline{n}} V_{h+1}^{\mathrm{ref},l_i}(s_{h+1}^{l_i}) \right)^2 - \frac{1}{\underline{n}}\left( \sum_{i=1}^{n} P_{s,a,h}V_{h+1}^{\mathrm{ref},l_i} \right)^2, \tag{39}$$

$$\chi_5 := \frac{1}{\underline{n}}\left( \sum_{i=1}^{\underline{n}} P_{s,a,h}V_{h+1}^{\mathrm{ref},l_i} \right)^2 - \sum_{i=1}^{\underline{n}} \left( P_{s,a,h}V_{h+1}^{\mathrm{ref},l_i} \right)^2. \tag{40}$$

By Azuma's inequality, we have $|\chi_3| \leqslant H^2\sqrt{2\underline{n}\iota}$ with probability at least $(1-p)$. We apply Azuma's inequality again to obtain that with probability at least $(1-p)$, it holds that

$$|\chi_4| = \frac{1}{\underline{n}}\left| \left( \sum_{i=1}^{\underline{n}} V_{h+1}^{\mathrm{ref},l_i}(s_{h+1}^{l_i}) \right)^2 - \left( \sum_{i=1}^{\underline{n}} P_{s,a,h}V_{h+1}^{\mathrm{ref},l_i} \right)^2 \right|$$

$$\leqslant 2H \cdot \left| \sum_{i=1}^{\underline{n}} V_{h+1}^{\mathrm{ref},l_i}(s_{h+1}^{l_i}) - \sum_{i=1}^{\underline{n}} P_{s,a,h}V_{h+1}^{\mathrm{ref},l_i} \right|$$

$$\leqslant 2H^2\sqrt{2\underline{n}\iota}. \tag{41}$$

On the other hand, we have that $\chi_5 \leqslant 0$ by Cauchy-Schwartz inequality. The proof then is completed by (37).

$\square$

Combing (34) with (36) we have

$$|\chi_1| \leqslant 2\sqrt{\frac{\nu^{\mathrm{ref}}\iota}{\underline{n}}} + \frac{5H\iota^{\frac{3}{4}}}{(\underline{n})^{\frac{3}{4}}} + \frac{2\sqrt{\iota}}{T\underline{n}} + \frac{2H\iota}{\underline{n}}. \tag{42}$$

We now bound $\sum_{i=1}^{\check{n}} \mathbb{V}(P_{s,a,h}, W_{h+1}^{\check{l}_i})$ for $|\chi_2|$. Define

$$\check{\nu} := \frac{\check{\sigma}}{\check{n}} - \left( \frac{\check{\mu}}{\check{n}} \right)^2.$$

Similarly to Lemma 12, we have that

**Lemma 13.** *With probability* $(1-2p)$, *it holds that*

$$\sum_{i=1}^{\check{n}} \mathbb{V}(P_{s,a,h}, W_{h+1}^{\check{l}_i}) \leqslant \check{n} \cdot \check{\nu} + 3H^2\sqrt{\check{n}\iota}. \tag{43}$$

Therefore, given (35), it holds with probability $(1 - 2p)$ that

$$|\chi_2| \leqslant 2\sqrt{\frac{\breve{\nu}\iota}{\breve{n}}} + \frac{5H\iota^{\frac{3}{4}}}{(\breve{n})^{\frac{3}{4}}} + \frac{2\sqrt{\iota}}{T\breve{n}} + \frac{2H\iota}{\breve{n}}. \tag{44}$$

Finally, combining (42), (44), and the definition of $\underline{b}$ with $[c_1, c_2, c_3] = [2, 2, 5]$, and collecting probabilities, we have that with probability at least $(1 - 2(H^2T^3 + 3))p$, it holds that

$$\underline{b} \geqslant |\chi_1| + |\chi_2|, \tag{45}$$

which means that $Q_h^{k+1}(s, a) \geqslant Q_h^*(s, a)$.

*For the second case,* by Hoeffding's inequality, with probability $(1 - p)$ it holds that

$$\begin{aligned}
Q_h^{k+1}(s, a) &= r_h(s, a) + \frac{\breve{v}}{\breve{n}} + \bar{b} \\
&\geqslant r_h(s, a) + \frac{1}{\breve{n}} \sum_{i=1}^{\breve{n}} V_{h+1}^*(s_{\breve{l}_i, h+1}) + 2\sqrt{\frac{H^2}{\breve{n}}\iota} \\
&\geqslant r_h(s, a) + P_{s,a,h}V_{h+1}^* \\
&= Q_h^*(s, a).
\end{aligned} \tag{46}$$

Combining the two cases, and via a union bound over all time steps, we prove the proposition.

## B.2 Proof of Lemma 5

First, by Hoeffding's inequality, for every $k$ and $h$, we have that

$$\mathbb{P}\left[ \left| \frac{1}{\breve{n}_h^k} \sum_{i=1}^{\breve{n}_h^k} V_{h+1}^*(s_{h+1}^{\breve{l}_i}) - P_{s_h^k, a_h^k, h}V_{h+1}^* \right| \leqslant \bar{b}_h^k \right] > 1 - p. \tag{47}$$

Now the whole proof will be conditioned on that (47) holds for every $k$ and $h$, which happens with probability at least $(1 - Tp)$. For every $k$ and $h$, we let $\delta_h^k := V_h^k(s_h^k) - V_h^*(s_h^k)$ (which aligns with the definition for $\delta_h^k$ in the proof of Theorem 1).

For any weight sequence $\{w^k\}_{k=1}^K$ such that $w^k \geqslant 0$, let $\|w\|_\infty = \max_{k=1}^K w^k$ and $\|w\|_1 = \sum_{k=1}^K w^k$. We will prove that

$$\sum_{k=1}^K w^k \delta_h^k \leqslant 240H^{\frac{5}{2}}\sqrt{\|w\|_\infty \cdot SA\|w\|_1 \iota} + 3SAH^3\|w\|_\infty. \tag{48}$$

Once we have established (48), we let $w^k = \mathbb{I}[\delta_h^k \geqslant \epsilon]$ and we have

$$\sum_{k=1}^K \mathbb{I}[\delta_h^k \geqslant \epsilon]\delta_h^k \leqslant 240H^{\frac{5}{2}}\sqrt{\|w\|_\infty \cdot SA\iota \sum_{k=1}^K \mathbb{I}[\delta_h^k \geqslant \epsilon]} + 3SAH^3\|w\|_\infty.$$

Note that $\|w\|_\infty$ is either 0 or 1. In either cases, we are able to derive that

$$\sum_{k=1}^K \mathbb{I}[\delta_h^k \geqslant \epsilon] \leqslant O(SAH^5\iota/\epsilon^2),$$

and concludes the proof of the lemma. Therefore, we only need to prove (48), and the rest of the proof is devoted to establishing (48).

By the update rule (9) and (10), that $V^k$ always upper bounds $V^*$ (conditioned on the successful event of Proposition 4), and that we have conditioned on (47), we have that

$$\begin{aligned}
\delta_h^k &= V_h^k(s_h^k) - V_h^*(s_h^k) \\
&\leqslant Q_h^k(s_h^k, a_h^k) - Q_h^*(s_h^k, a_h^k)
\end{aligned}$$

$$\leqslant \mathbb{I}[n_h^k = 0]H + \left(\bar{b}_h^k + \frac{1}{\check{n}_h^k}\sum_{i=1}^{\check{n}_h^k}V_{h+1}^{\check{l}_i}(s_{h+1}^{\check{l}_i}) - P_{s_h^k,a_h^k,h}V_{h+1}^*\right)$$

$$\leqslant \mathbb{I}[n_h^k = 0]H + \left(2\bar{b}_h^k + \frac{1}{\check{n}_h^k}\sum_{i=1}^{\check{n}_h^k}(V_{h+1}^{\check{l}_i}(s_{h+1}^{\check{l}_i}) - V_{h+1}^*(s_{h+1}^{\check{l}_i}))\right)$$

$$= \mathbb{I}[n_h^k = 0]H + \left(2\bar{b}_h^k + \frac{1}{\check{n}_h^k}\sum_{i=1}^{\check{n}_h^k}\delta_{h+1}^{\check{l}_i}\right). \tag{49}$$

Using the similar trick we do for (15) and (16), we have

$$\sum_{k=1}^{K}\frac{w^k}{\check{n}_h^k}\sum_{i=1}^{\check{n}_h^k}\delta_{h+1}^{\check{l}_i} = \sum_{j=1}^{K}\frac{w^j}{\check{n}_h^j}\sum_{i=1}^{\check{n}_h^j}\delta_{h+1}^{\check{l}_{h,i}^j}$$

$$= \sum_{j=1}^{K}\frac{w^j}{\check{n}_h^j}\sum_{k=1}^{K}\delta_{h+1}^k\sum_{i=1}^{\check{n}_h^j}\mathbb{I}[k = \check{l}_{h,i}^j] = \sum_{k=1}^{K}\delta_{h+1}^k\sum_{j=1}^{K}\frac{w^j}{\check{n}_h^j}\sum_{i=1}^{\check{n}_h^j}\mathbb{I}[k = \check{l}_{h,i}^j], \tag{50}$$

where if we let

$$\tilde{w}^k = \sum_{j=1}^{K}\frac{w^j}{\check{n}_h^j}\sum_{i=1}^{\check{n}_h^j}\mathbb{I}[k = \check{l}_{h,i}^j], \tag{51}$$

we have that

$$\|\tilde{w}\|_\infty = \max_k \tilde{w}^k \leqslant (1 + \frac{1}{H})\|w\|_\infty, \qquad \text{and} \qquad \|\tilde{w}\|_1 = \sum_k \tilde{w}^k = \sum_k w^k = \|w\|_1. \tag{52}$$

Therefore, combining (49), (50), and (51), and plugging them into $\sum_k w^k\delta_h^k$, we have that

$$\sum_k w^k\delta_h^k \leqslant 2\sum_k w^k\bar{b}_h^k + \sum_k \tilde{w}^k\delta_{h+1}^k + H\sum_k w^k\mathbb{I}[n_h^k = 0]$$

$$\leqslant 2\sum_k w^k\bar{b}_h^k + \sum_k \tilde{w}^k\delta_{h+1}^k + SAH^2\|w\|_\infty, \tag{53}$$

We now bound the first term of (53). Define $\mathfrak{w}(s,a,j) := \sum_{k=1}^{K}w^k\mathbb{I}[\check{n}_h^k = e_j, (s_h^k, a_h^k) = (s,a)]$ and $\mathfrak{w}(s,a) := \sum_{j\geqslant 1}\mathfrak{w}(s,a,j)$. We have $\mathfrak{w}(s,a,j) \leqslant \|w\|_\infty(1 + \frac{1}{H})e_j$ and $\sum_{s,a}\mathfrak{w}(s,a) = \sum_k w^k$. We then have

$$\sum_k w^k\bar{b}_h^k = \sum_k 2\sqrt{H^2\iota}w^k\sqrt{\frac{1}{\check{n}_h^k}}$$

$$= 2\sqrt{H^2\iota}\sum_{s,a,j}\sqrt{\frac{1}{e_j}}\sum_{k=1}^{K}w^k\mathbb{I}[\check{n}_h^k = e_j, (s_h^k, a_h^k) = (s,a)] = 2\sqrt{H^2\iota}\sum_{s,a}\sum_{j\geqslant 1}\mathfrak{w}(s,a,j)\sqrt{\frac{1}{e_j}}.$$

We fix $(s,a)$ and consider the sum $\sum_{j\geqslant 1}\mathfrak{w}(s,a,j)\sqrt{\frac{1}{e_j}}$. Notice that $\sqrt{1/e_j}$ is monotonically decreasing in $j$. Given that $\sum_{j\geqslant 0}\mathfrak{w}(s,a,j) = \mathfrak{w}(s,a)$ is fixed, by rearrangement inequality we have that

$$\sum_{j\geqslant 1}\mathfrak{w}(s,a,j)\sqrt{\frac{1}{e_j}} \leqslant \sum_{j\geqslant 1}\sqrt{\frac{1}{e_j}}\cdot\|w\|_\infty(1 + \frac{1}{H})e_j\cdot\mathbb{I}\left[\sum_{i=1}^{j-1}\|w\|_\infty(1 + \frac{1}{H})e_i \leqslant \mathfrak{w}(s,a)\right]$$

$$= \|w\|_\infty(1 + \frac{1}{H})\sum_j\sqrt{e_j}\cdot\mathbb{I}\left[\sum_{i=1}^{j-1}\|w\|_\infty e_i \leqslant \mathfrak{w}(s,a)\right]$$

$$\leqslant 10(1 + \frac{1}{H})\sqrt{\|w\|_\infty H\cdot\mathfrak{w}(s,a)}.$$

Therefore, by Cauchy-Schwartz, we have that

$$\sum_k w^k \bar{b}_h^k \leqslant 2\sqrt{H^2 \iota} \sum_{s,a} 10(1 + \frac{1}{H})\sqrt{\|w\|_\infty H}\sqrt{\mathfrak{w}(s,a)} \leqslant 20\sqrt{H^2 \iota}(1 + \frac{1}{H})\sqrt{\|w\|_\infty \cdot SAH\|w\|_1}. \tag{54}$$

Combining (53) and (54), we have that

$$\sum_k w^k \delta_h^k \leqslant 80H\sqrt{\|w\|_\infty \cdot SAH\|w\|_1 \iota} + SAH^2\|w\|_\infty + \sum_k \tilde{w}^k \delta_{h+1}^k. \tag{55}$$

With (55) and (52) in hand, applying induction on $h$ with the base case that $h = H$, one may deduce that

$$\sum_k w^k \delta_h^k \leqslant (1 + 1/H)^H \cdot H \cdot \left(80H\sqrt{\|w\|_\infty \cdot SAH\|w\|_1 \iota} + SAH^2\|w\|_\infty\right)$$

$$\leqslant 240H^2\sqrt{\|w\|_\infty \cdot SAH\|w\|_1 \iota} + 3SAH^3\|w\|_\infty.$$

## B.3 Proof of Lemma 7

The entire proof is conditioned on the successful events of Proposition 4 and Lemma 5 which happen with probability at least $(1 - 2T(H^2T^3 + 5)p)$. For convenience, we define $\lambda_h^k$ as $\lambda_h^k(s) = \mathbb{I}\left[n_h^k(s) < N_0\right]$ for all state $s$ and all $k$ and $h$.

By the definition of $\Lambda_{h+1}^k$, we have that $\sum_{h=1}^H \sum_{k=1}^K (1 + \frac{1}{H})^{h-1} \Lambda_{h+1}^k$ by the definition that

$$\sum_{h=1}^H \sum_{k=1}^K (1 + \frac{1}{H})^{h-1} \Lambda_{h+1}^k = \sum_{h=1}^H \sum_{k=1}^K (1 + \frac{1}{H})^{h-1} \psi_{h+1}^k + \sum_{h=1}^H \sum_{k=1}^K (1 + \frac{1}{H})^{h-1} \xi_{h+1}^k$$

$$+ \sum_{h=1}^H \sum_{k=1}^K (1 + \frac{1}{H})^{h-1} \phi_{h+1}^k + 2\sum_{h=1}^H \sum_{k=1}^K (1 + \frac{1}{H})^{h-1} b_h^k. \tag{56}$$

We will bound the four terms separately.

### B.3.1 The $\psi_{h+1}^k$ Term

**Lemma 14.** *With probability at least $(1 - p)$, it holds that*

$$\sum_{h=1}^H \sum_{k=1}^K (1 + \frac{1}{H})^{h-1} \psi_{h+1}^k \leqslant O(\log(T)) \cdot (H^2 SN_0 + H\sqrt{T\iota}).$$

*Proof.* Because $\psi_{h+1}^k$ is always non-negative, we have that with probability $(1 - p)$ it holds that

$$\sum_{k=1}^K \sum_{h=1}^H (1 + \frac{1}{H})^{h-1} \psi_{h+1}^k$$

$$\leqslant 3\sum_{k=1}^K \sum_{h=1}^H \psi_{h+1}^k$$

$$= 3\sum_{k=1}^K \sum_{h=1}^H \frac{1}{n_h^k} \sum_{i=1}^{n_h^k} P_{s_h^k, a_h^k, h}(V_{h+1}^{\text{ref},l_i} - V_{h+1}^{\text{REF}})$$

$$\leqslant 3H\sum_{k=1}^K \sum_{h=1}^H \frac{1}{n_h^k} \sum_{i=1}^{n_h^k} P_{s_h^k, a_h^k, h}\lambda_{h+1}^{l_i}$$

$$\leqslant 3H\sum_{h=1}^H \sum_{j=1}^K \sum_{k=1}^K P_{s_h^k, a_h^k, h}\lambda_{h+1}^j \cdot \frac{1}{n_h^k} \sum_{i=1}^{n_h^k} \mathbb{I}[l_{h,i}^k = j]$$

$$\leqslant 3H\sum_{h=1}^H \sum_{j=1}^K P_{s_h^j, a_h^j, h}\lambda_{h+1}^j \cdot \sum_{k=1}^K \frac{1}{n_h^k} \sum_{i=1}^{n_h^k} \mathbb{I}[l_{h,i}^k = j] \tag{57}$$

$$\leqslant 6\big(\log(T)+1\big)\cdot H\sum_{h=1}^{H}\sum_{k=1}^{K}P_{s_h^k,a_h^k,h}\lambda_{h+1}^k \tag{58}$$

$$= 6\big(\log(T)+1\big)\cdot H\Big(\sum_{k=1}^{K}\sum_{h=1}^{H}\lambda_{h+1}^k(s_{h+1}^k)+\sum_{k=1}^{K}\sum_{h=1}^{H}(P_{s_h^k,a_h^k,h}-\mathbf{1}_{s_{h+1}^k})\lambda_{h+1}^k\Big)$$

$$\leqslant 6\big(\log(T)+1\big)\cdot H\Big(HSN_0+\sum_{k=1}^{K}\sum_{h=1}^{H}(P_{s_h^k,a_h^k,h}-\mathbf{1}_{s_{h+1}^k})\lambda_{h+1}^k\Big)$$

$$\leqslant 6\big(\log(T)+1\big)\cdot H\Big(HSN_0+2\sqrt{T\iota}\Big). \tag{59}$$

Here, Inequality (57) is because $\frac{1}{n_h^k}\sum_{i=1}^{n_h^k}\mathbb{I}[l_{h,i}^k=j]\neq 0$ only if $(s_h^k,a_h^k)=(s_h^j,a_h^j)$. Inequality (58) is because

$$\sum_{k=1}^{K}\frac{1}{n_h^k}\sum_{i=1}^{n_h^k}\mathbb{I}\big[l_{h,i}^k=j\big]\leqslant\sum_{z:j\leqslant\sum_{i=1}^{z-1}e_i\leqslant T}\frac{e_z}{\sum_{i=1}^{z-1}e_i}\leqslant 2(\log(T)+1).$$

Inequality (59) holds with probability $(1-p)$ due to Azuma's inequality. □

### B.3.2 The $\xi_{h+1}^k$ Term

**Lemma 15.** *With probability at least $(1-(T+1)p)$, it holds that*

$$\sum_{k=1}^{K}\sum_{h=1}^{H}(1+\frac{1}{H})^{h-1}\xi_{h+1}^k\leqslant O(H\sqrt{SAT\iota}).$$

*Proof.* We have that

$$\sum_{k=1}^{K}\sum_{h=1}^{H}(1+\frac{1}{H})^{h-1}\xi_{h+1}^k=\sum_{k=1}^{K}\sum_{h=1}^{H}(1+\frac{1}{H})^{h-1}\Big(\frac{1}{\check{n}_h^k}\sum_{i=1}^{\check{n}_h^k}(P_{s_h^k,a_h^k,h}-\mathbf{1}_{s_{h+1}^{\check{l}_i}})(V_{h+1}^{\check{l}_i}-V_{h+1}^*)\Big)$$

$$=\sum_{k=1}^{K}\sum_{h=1}^{H}\sum_{j=1}^{K}(1+\frac{1}{H})^{h-1}\Big(\frac{1}{\check{n}_h^k}\sum_{i=1}^{\check{n}_h^k}(P_{s_h^k,a_h^k,h}-\mathbf{1}_{s_{h+1}^j})(V_{h+1}^j-V_{h+1}^*)\cdot\mathbb{I}[\check{l}_{h,i}^k=j]\Big).$$

Note that in the expression above $\check{l}_{h,i}^k=j$ if and only if $(s_h^k,a_h^k)=(s_h^j,s_h^j)$. Therefore, we have

$$\sum_{k=1}^{K}\sum_{h=1}^{H}(1+\frac{1}{H})^{h-1}\xi_{h+1}^k$$

$$=\sum_{k=1}^{K}\sum_{h=1}^{H}\sum_{j=1}^{K}(1+\frac{1}{H})^{h-1}\Big(\frac{1}{\check{n}_h^k}\sum_{i=1}^{\check{n}_h^k}(P_{s_h^j,a_h^j,h}-\mathbf{1}_{s_{h+1}^j})(V_{h+1}^j-V_{h+1}^*)\cdot\mathbb{I}[\check{l}_{h,i}^k=j]\Big)$$

$$=\sum_{h=1}^{H}\sum_{j=1}^{K}(1+\frac{1}{H})^{h-1}(P_{s_h^j,a_h^j,h}-\mathbf{1}_{s_{h+1}^j})(V_{h+1}^j-V_{h+1}^*)\cdot\sum_{k=1}^{K}\frac{1}{\check{n}_h^k}\sum_{i=1}^{\check{n}_h^k}\mathbb{I}[\check{l}_{h,i}^k=j]$$

$$=\sum_{k=1}^{K}\sum_{h=1}^{H}\theta_{h+1}^k(P_{s_h^k,a_h^k,h}-\mathbf{1}_{s_{h+1}^k})(V_{h+1}^k-V_{h+1}^*), \tag{60}$$

where we define $\theta_{h+1}^j:=(1+\frac{1}{H})^{h-1}\sum_{k=1}^{K}\big(\frac{1}{\check{n}_h^k}\sum_{i=1}^{\check{n}_h^k}\mathbb{I}[\check{l}_{h,i}^k=j]\big)$.

For $(j,h)\in[K]\times[H]$, let $x_h^j$ be the number of elements in current stage with respect to $(s_h^j,a_h^j,h)$ and $\tilde{\theta}_{h+1}^j:=(1+\frac{1}{H})^{h-1}\frac{\lfloor(1+\frac{1}{H})x_h^j\rfloor}{x_h^j}\leqslant 3$. Define $\mathcal{K}=\{(k,h):\theta_{h+1}^k=\tilde{\theta}_{h+1}^k\}$. Note that if $k$ is before the second last stage (before the final episode $K$) of the triple $(s_h^k,a_h^k,h)$, then we have

$\theta_{h+1}^k = \tilde{\theta}_{h+1}^k$ and $(k,h) \in \mathcal{K}$. Given that $(k,h) \in \mathcal{K}$, $s_{h+1}^k$ still follows the transition distribution $P_{s_h^k, a_h^k, h}$.

Let $\mathcal{K}_h^\perp(s,a) = \{k : (s_h^k, a_h^k) = (s,a), k \text{ is in the second last stage of } (s,a,h)\}$. Note that for two different episodes $j, k$, if $(s_h^k, a_h^k) = (s_h^j, a_h^j)$ and $j, k$ are in the same stage of $(s_h^k, a_h^k, h)$, then $\theta_{h+1}^k = \theta_{h+1}^j$ and $\tilde{\theta}_{h+1}^k = \tilde{\theta}_{h+1}^j$. Let $\theta_{h+1}(s,a)$ and $\tilde{\theta}_{h+1}(s,a)$ to denote $\theta_{h+1}^k$ and $\tilde{\theta}_{h+1}^k$ respectively for some $k \in \mathcal{K}_h^\perp(s,a)$.

We rewrite as

$$\sum_{k=1}^K \sum_{h=1}^H (1 + \frac{1}{H})^{h-1} \xi_{h+1}^k$$
$$= \sum_{(k,h)} \tilde{\theta}_{h+1}^k (P_{s_h^k, a_h^k, h} - \mathbf{1}_{s_{h+1}^k})(V_{h+1}^k - V_{h+1}^*) + \sum_{(k,h) \in \overline{\mathcal{K}}} (\theta_{h+1}^k - \tilde{\theta}_{h+1}^k)(P_{s_h^k, a_h^k, h} - \mathbf{1}_{s_{h+1}^k})(V_{h+1}^k - V_{h+1}^*).$$
(61)

Because $\tilde{\theta}_{h+1}^k$ is independent from $s_{h+1}^k$, by Azuma's inequality, we have with probability $(1-p)$, it holds that

$$\sum_{(k,h)} \tilde{\theta}_{h+1}^k (P_{s_h^k, a_h^k, h} - \mathbf{1}_{s_{h+1}^k})(V_{h+1}^k - V_{h+1}^*) \leqslant 6\sqrt{TH^2 \iota}. \tag{62}$$

For the second term in (61), we have that

$$\sum_{(k,h) \in \overline{\mathcal{K}}} (\theta_{h+1}^k - \tilde{\theta}_{h+1}^k)(P_{s_h^k, a_h^k, h} - \mathbf{1}_{s_{h+1}^k})(V_{h+1}^k - V_{h+1}^*)$$
$$= \sum_{s,a,h} \sum_{k:(k,h) \in \overline{\mathcal{K}}} \mathbb{I}[(s_h^k, a_h^k) = (s,a)](\theta_{h+1}^k - \tilde{\theta}_{h+1}^k)(P_{s_h^k, a_h^k, h} - \mathbf{1}_{s_{h+1}^k})(V_{h+1}^k - V_{h+1}^*)$$
$$= \sum_{s,a,h} (\theta_{h+1}(s,a) - \tilde{\theta}_{h+1}(s,a)) \sum_{k \in \mathcal{K}_h^\perp(s,a)} (P_{s_h^k, a_h^k, h} - \mathbf{1}_{s_{h+1}^k})(V_{h+1}^k - V_{h+1}^*)$$
$$\leqslant \sum_{s,a,h} O(H)\sqrt{|\mathcal{K}_h^\perp(s,a)| \iota} \tag{63}$$
$$= \sum_{(s,a,h)} O(H) \cdot \sqrt{\check{N}_h^{K+1}(s,a) \iota} \tag{}$$
$$\leqslant O(H) \cdot \sqrt{SAH\iota \sum_{(s,a,h)} \check{N}_h^{K+1}(s,a)} \tag{64}$$
$$\leqslant O(H) \cdot \sqrt{SAH\iota \cdot (T/H)}. \tag{65}$$

Here, (63) happens with probability $(1 - Tp)$ because of Azuma's inequality and a union bound over all times steps in $\overline{\mathcal{K}}$. (64) is due to Cauchy-Schwartz, and (65) is because the length of the last two stages for each $(s,a,h)$ triple is only $O(1/H)$ fraction of the total number of visits.

Combining (61), (62), (65), and collecting probabilities, we prove the desired result. $\qquad \square$

### B.3.3 The $\phi_{h+1}^k$ Term

**Lemma 16.** *With probability $(1-p)$, it holds that*

$$\sum_{k=1}^K \sum_{h=1}^H (1 + \frac{1}{H})^{h-1} \phi_{k+1}^k = \sum_{k=1}^K \sum_{h=1}^H (1 + \frac{1}{H})^{h-1}(P_{s_h^k, a_h^k, h} - \boldsymbol{1}_{s_{h+1}^k})(V_{h+1}^* - V_{h+1}^{\pi_k}) \leqslant O(\sqrt{H^2 T\iota}).$$

*Proof.* The lemma follows easily from Azuma's inequality. $\qquad \square$

### B.3.4 The $b_h^k$ Term

**Lemma 17.** *With probability* $(1 - 9p)$, *it holds that*

$$\sum_{k=1}^{K} \sum_{h=1}^{H} (1 + \frac{1}{H})^{h-1} b_h^k \leqslant O\Big(\sqrt{SAH^2T\iota} + \sqrt{SAH^2\beta T\iota} + SAH^3\sqrt{SN_0\iota}\log(T)$$

$$+ \sqrt{SAH^3\beta^2T\iota} + (SA\iota)^{\frac{3}{4}}H^{\frac{5}{2}}T^{\frac{1}{4}}\Big).$$

*Proof.* Define $\nu_h^{\text{ref},k} = \frac{\sigma_h^{\text{ref},k}}{n_h^k} - (\frac{\mu_h^{\text{ref},k}}{n_h^k})^2$ and $\check{\nu}_h^k = \frac{\check{\sigma}_h^k}{\check{n}_h^k} - (\frac{\check{\mu}_h^k}{\check{n}_h^k})^2$. Since $b_h^k$ is non-negative, we have that

$$2\sum_{h=1}^{H}\sum_{k=1}^{K}(1 + \frac{1}{H})^{h-1}b_h^k$$

$$\leqslant 6\sum_{h=1}^{H}\sum_{k=1}^{K}\Big(c_1\sqrt{\frac{\nu_h^{\text{ref},k}}{n_h^k}\iota} + c_2\sqrt{\frac{\check{\nu}_h^k}{\check{n}_h^k}\iota} + c_3\big(\frac{H\iota}{n_h^k} + \frac{H\iota}{\check{n}_h^k} + \frac{H\iota^{\frac{3}{4}}}{(n_h^k)^{\frac{3}{4}}} + \frac{H\iota^{\frac{3}{4}}}{(\check{n}_h^k)^{\frac{3}{4}}}\big)\Big) \quad (66)$$

$$\leqslant O\Big(\sum_{h=1}^{H}\sum_{k=1}^{K}(\sqrt{\frac{\nu_h^{\text{ref},k}}{n_h^k}\iota} + \sqrt{\frac{\check{\nu}_h^k}{\check{n}_h^k}\iota})\Big) + O\Big(SAH^3\log(T)\iota + (SA\iota)^{\frac{3}{4}}H^{\frac{5}{2}}T^{\frac{1}{4}}\Big). \quad (67)$$

Inequality (67) is due to Lemma 11 with $\alpha = \frac{3}{4}$ and $\alpha = 1$. Now we only need to analyze the first term in (67).

We first present an upper bound for $\nu_h^{\text{ref},k}$. Recall that $\mathbb{V}(x, y) = x^\top(y^2) - (x^\top y)^2$.

**Lemma 18.** *With probability* $(1 - 4p)$, *it holds that*

$$\nu_h^{\text{ref},k} - \mathbb{V}(P_{s_h^k, a_h^k, h}, V_{h+1}^*) \leqslant 4H\beta + \frac{6H^2SN_0}{n_h^k} + 14H^2\sqrt{\frac{\iota}{n_h^k}}.$$

*Proof.* We prove by first bounding $\nu_h^{\text{ref},k} - \frac{1}{n_h^k}\sum_{i=1}^{n_h^k}\mathbb{V}(P_{s_h^k, a_h^k, h}, V_{h+1}^{\text{ref},l_i})$. Recall that by (37),

$$\nu_h^{\text{ref},k} - \frac{1}{n_h^k}\sum_{i=1}^{n_h^k}\mathbb{V}(P_{s_h^k, a_h^k, h}, V_{h+1}^{\text{ref},l_i}) = -\frac{1}{n_h^k}(\chi_6 + \chi_7 + \chi_8),$$

where

$$\chi_6 := \sum_{i=1}^{n_h^k}\Big(P_{s,a,h}(V_{h+1}^{\text{ref},l_i})^2 - (V_{h+1}^{\text{ref},l_i}(s_{h+1}^{l_i}))^2\Big), \quad (68)$$

$$\chi_7 := \frac{1}{n_h^k}\Big(\sum_{i=1}^{n_h^k}V_{h+1}^{\text{ref},l_i}(s_{h+1}^{l_i})\Big)^2 - \frac{1}{n_h^k}\Big(\sum_{i=1}^{n_h^k}P_{s,a,h}V_{h+1}^{\text{ref},l_i}\Big)^2, \quad (69)$$

$$\chi_8 := \frac{1}{n_h^k}\Big(\sum_{i=1}^{n_h^k}P_{s,a,h}V_{h+1}^{\text{ref},l_i}\Big)^2 - \sum_{i=1}^{n_h^k}\Big(P_{s,a,h}V_{h+1}^{\text{ref},l_i}\Big)^2. \quad (70)$$

By Azuma's inequality, with probability $(1 - 2p)$ it holds that

$$|\chi_6| \leqslant H^2\sqrt{2n_h^k\iota},$$

$$|\chi_7| \leqslant 2H|\sum_{i=1}^{n_h^k}V_{h+1}^{\text{ref},l_i}(s_{h+1}^{l_i}) - \sum_{i=1}^{n_h^k}P_{s_h^k, a_h^k, h}V^{\text{ref},l_i}| \leqslant 2H^2\sqrt{2n_h^k\iota}.$$

It left us to handle $-\chi_8$. By Azuma's inequality and the fact that $V^{\mathrm{ref},k} \geqslant V^{\mathrm{REF}}$ for any $k$, with probability $(1-p)$ it holds that

$$
\begin{aligned}
-\chi_8 &= \sum_{i=1}^{n_h^k} \left(P_{s_h^k,a_h^k,h} V_{h+1}^{\mathrm{ref},l_i}\right)^2 - \frac{1}{n_h^k}\left(\sum_{i=1}^{n_h^k} P_{s_h^k,a_h^k,h} V_{h+1}^{\mathrm{ref},l_i}\right)^2 \\
&\leqslant \sum_{i=1}^{n_h^k} \left(P_{s_h^k,a_h^k,h} V_{h+1}^{\mathrm{ref},l_i}\right)^2 - \frac{1}{n_h^k}\left(\sum_{i=1}^{n_h^k} P_{s_h^k,a_h^k,h} V_{h+1}^{\mathrm{REF}}\right)^2 \\
&= \sum_{i=1}^{n_h^k} \left(\left(P_{s_h^k,a_h^k,h} V_{h+1}^{\mathrm{ref},l_i}\right)^2 - (P_{s_h^k,a_h^k,h} V_{h+1}^{\mathrm{REF}})^2\right) \\
&\leqslant 2H^2 \sum_{i=1}^{n_h^k} P_{s_h^k,a_h^k,h} \lambda_{h+1}^{l_i} \\
&= 2H^2 \left(\sum_{i=1}^{n_h^k} \lambda_{h+1}^{l_i}(s_{h+1}^{l_i}) + \sum_{i=1}^{n_h^k}(P_{s_h^k,a_h^k,h} - \mathbf{1}_{s_{h+1}^{l_i}})\lambda_{h+1}^{l_i}\right) \\
&\leqslant 2H^2 S N_0 + 3H^2\sqrt{n_h^k \iota}.
\end{aligned}
\tag{71}
$$

Then we obtain that

$$
\nu_h^{\mathrm{ref},k} - \frac{1}{n_h^k}\sum_{i=1}^{n_h^k}\mathbb{V}(P_{s_h^k,a_h^k,h}, V_{h+1}^{\mathrm{ref},l_i}) \leqslant 8H^2\sqrt{\frac{\iota}{n_h^k}} + \frac{2H^2 S N_0}{n_h^k}.
\tag{72}
$$

When (72) holds, we have that with probability $(1-p)$,

$$
\begin{aligned}
&\nu_h^{\mathrm{ref},k} - \mathbb{V}(P_{s_h^k,a_h^k,h}, V_{h+1}^*) \\
&= \frac{1}{n_h^k}\sum_{i=1}^{n_h^k}\left(\mathbb{V}(P_{s_h^k,a_h^k,h}, V_{h+1}^{\mathrm{ref},l_i}) - \mathbb{V}(P_{s_h^k,a_h^k,h}, V_{h+1}^*)\right) + \left(\nu_h^{\mathrm{ref},k} - \frac{1}{n_h^k}\sum_{i=1}^{n_h^k}\mathbb{V}(P_{s_h^k,a_h^k,h}, V_{h+1}^{\mathrm{ref},l_i})\right) \\
&\leqslant \frac{1}{n_h^k}\sum_{i=1}^{n_h^k}\left(\mathbb{V}(P_{s_h^k,a_h^k,h}, V_{h+1}^{\mathrm{ref},l_i}) - \mathbb{V}(P_{s_h^k,a_h^k,h}, V_{h+1}^*)\right) + 8H^2\sqrt{\frac{\iota}{n_h^k}} + \frac{2H^2 S N_0}{n_h^k} \\
&\leqslant \frac{4H}{n_h^k}\sum_{i=1}^{n_h^k} P_{s_h^k,a_h^k,h}(V_{h+1}^{\mathrm{ref},l_i} - V_{h+1}^*) + 8H^2\sqrt{\frac{\iota}{n_h^k}} + \frac{2H^2 S N_0}{n_h^k} \\
&\leqslant \frac{4H}{n_h^k}\sum_{i=1}^{n_h^k}(V_{h+1}^{\mathrm{ref},l_i}(s_{h+1}^{l_i}) - V_{h+1}^*(s_{h+1}^{l_i})) + \frac{4H}{n_h^k}\sum_{i=1}^{n_h^k}(P_{s_h^k,a_h^k,h} - \mathbf{1}_{s_{h+1}^{l_i}})(V_{h+1}^{\mathrm{ref},l_i} - V_{h+1}^*) \\
&\quad + 8H^2\sqrt{\frac{\iota}{n_h^k}} + \frac{2H^2 S N_0}{n_h^k} \\
&\leqslant \frac{4H}{n_h^k}\sum_{i=1}^{n_h^k}(V_{h+1}^{\mathrm{ref},l_i}(s_{h+1}^{l_i}) - V_{h+1}^*(s_{h+1}^{l_i})) + 14H^2\sqrt{\frac{\iota}{n_h^k}} + \frac{2H^2 S N_0}{n_h^k} \\
&\leqslant \frac{4H}{n_h^k}\sum_{i=1}^{n_h^k}(H\lambda_{h+1}^{l_i}(s_{h+1}^{l_i}) + \beta) + 14H^2\sqrt{\frac{\iota}{n_h^k}} + \frac{2H^2 S N_0}{n_h^k} \\
&\leqslant 4H\beta + \frac{6H^2 S N_0}{n_h^k} + 14H^2\sqrt{\frac{\iota}{n_h^k}},
\end{aligned}
$$

(73) appears at the fifth display, (74) at the sixth display.

where Inequality (73) holds with probability $(1-p)$ by Azuma's inequality and (74) holds by Corollary 6 (and note that the whole proof is conditioned on the successful events of Proposition 4 and Lemma 5). $\square$

We will also prove the following bound of the total variance.

**Lemma 19.** *With probability* $(1 - 2p)$, *it holds that*

$$\sum_{s,a,h} N_h^{K+1}(s,a)\mathbb{V}(P_{s,a,h}, V_{h+1}^*) \leqslant 2TH + 3\sqrt{2H^4T\iota}. \tag{75}$$

*Proof.* By direct calculation, with probability $(1 - 2p)$, it holds that

$$\sum_{s,a,h} N_h^{K+1}(s,a)\mathbb{V}(P_{s,a,h}, V_{h+1}^*)$$

$$= \sum_{k=1}^{K}\sum_{h=1}^{H}\mathbb{V}(P_{s_h^k,a_h^k,h}, V_{h+1}^*)$$

$$= \sum_{k=1}^{K}\sum_{h=1}^{K}\left(P_{s_h^k,a_h^k,h}(V_{h+1}^*)^2 - (P_{s_h^k,a_h^k,h}V_{h+1}^*)^2\right)$$

$$\leqslant \sum_{k=1}^{K}\sum_{h=1}^{H}\left(P_{s_h^k,a_h^k,h}(V_{h+1}^*)^2 - (V_h^*(s_h^k))^2\right) + 2H\sum_{k=1}^{K}\sum_{h=1}^{H}|V_h^*(s_h^k) - P_{s_h^k,a_h^k,h}V_{h+1}^*|$$

$$\leqslant \sqrt{2TH^4\iota} + 2H\sum_{k=1}^{K}\sum_{h=1}^{H}|V_h^*(s_h^k) - P_{s_h^k,a_h^k,h}V_{h+1}^*| \tag{76}$$

$$= \sqrt{2TH^4\iota} + 2H\sum_{k=1}^{K}\left(V_1^*(s_1^k) + \sum_{h=1}^{H}(V_h^*(s_{h+1}^k) - P_{s_h^k,a_h^k,h}V_{h+1}^*))\right) \tag{77}$$

$$\leqslant \sqrt{2TH^4\iota} + 2TH + 2H^2\sqrt{2T\iota} \tag{78}$$

$$\leqslant 2TH + 3H^2\sqrt{2T\iota},$$

where Inequality (76) holds with probability $(1 - p)$ by Azuma's inequality, Equation (77) holds with the fact that $V_h^*(s) - P_{s,a,h}V_{h+1}^* \geqslant V_h^*(s) - Q_h^*(s,a) \geqslant 0$ for any $s, a, h$ and Inequality (78) holds with probability $(1 - p)$ by Azuma's inequality. $\qquad\square$

Combining Lemma 11, Lemma 18, and Lemma 19, we have that with probability $(1 - 7p)$,

$$\sum_{h=1}^{H}\sum_{k=1}^{K}\sqrt{\frac{\nu_h^{\text{ref},k}}{n_h^k}\iota} \leqslant \sum_{h=1}^{H}\sum_{k=1}^{K}\sqrt{\frac{\mathbb{V}(P_{s_h^k,a_h^k,h}, V_{h+1}^*)}{n_h^k}\iota} + \sum_{h=1}^{H}\sum_{k=1}^{K}\sqrt{\left(\frac{4H\beta}{n_h^k} + \frac{6H^2SN_0}{(n_h^k)^2} + 14H^2\frac{\sqrt{\iota}}{(n_h^k)^{\frac{3}{2}}}\right)\iota}$$

$$\leqslant O\left(\sum_{s,a,h}\sqrt{N_h^{K+1}(s,a)\mathbb{V}(P_{s,a,h}, V_{h+1}^*)\iota}\right.$$

$$\left. + \sum_{s,a,h}\sqrt{N_h^{K+1}(s,a)H\beta\iota} + SAH^2\sqrt{SN_0\iota}\log(T) + (SA\iota)^{\frac{3}{4}}H^{\frac{7}{4}}T^{\frac{1}{4}}\right)$$

$$\leqslant O\left(\sqrt{SAH^2T\iota} + \sqrt{SAH^2\beta T\iota} + SAH^2\sqrt{SN_0\iota}\log(T) + (SA\iota)^{\frac{3}{4}}H^{\frac{7}{4}}T^{\frac{1}{4}}\right). \tag{79}$$

We now bound $\check{\nu}_h^k$. By Corollary 6 (and that the whole proof is conditioned on the successful events of Proposition 4 and Lemma 5), we have that

$$\check{\nu}_h^k \leqslant \frac{1}{\check{n}_h^k}\sum_{i=1}^{\check{n}_h^k}\left(V_{h+1}^{\text{ref},\check{l}_i}(s_{h+1}^{\check{l}_i}) - V_{h+1}^*(s_{h+1}^{\check{l}_i})\right)^2$$

$$\leqslant \frac{1}{\check{n}_h^k}\sum_{i=1}^{\check{n}_h^k}(H^2\lambda_{h+1}^{\check{l}_i}(s_{h+1}^{\check{l}_i}) + \beta^2)$$

$$\leqslant \frac{1}{\check{n}_h^k}H^2SN_0 + \beta^2. \tag{80}$$

By Lemma 11, we obtain that

$$\sum_{h=1}^{H}\sum_{k=1}^{K}\sqrt{\frac{\check{\nu}_h^k}{\check{n}_h^k}\iota} \leqslant \sum_{h=1}^{H}\sum_{k=1}^{K}\left(\sqrt{\frac{\beta^2}{\check{n}_h^k}\iota} + \frac{\sqrt{H^2SN_0\iota}}{\check{n}_h^k}\right) \leqslant O\left(\sqrt{SAH^3\beta^2T\iota} + SAH^3\sqrt{SN_0\iota}\log(T)\right).$$
(81)

The proof is completed by combining (67), (79), and (81).

$\square$

### B.3.5 Putting Everything Together

Recall that $\beta = \frac{1}{\sqrt{H}}$, and $N_0 = \frac{c_4 SAH^5\iota}{\beta^2} = O(SAH^6\iota)$. Combining (56), Lemma 14, Lemma 15, Lemma 16 and Lemma 17, we conclude that with probability at least $(1 - O(H^2T^4p))$,

$$\begin{aligned}
&\sum_{k=1}^{K}\sum_{h=1}^{K}\Lambda_{h+1}^k \\
&\leqslant O(\log(T)) \cdot (H^2SN_0 + H\sqrt{T\iota}) + O(H^2\sqrt{SAT\iota}) + O(\sqrt{H^2T\iota}) \\
&\quad + O\Big(\sqrt{SAH^2T\iota} + \sqrt{SAH^2\beta T\iota} + SAH^3\sqrt{SN_0\iota}\log(T) \\
&\qquad + \sqrt{SAH^3\beta^2T\iota} + (SA\iota)^{\frac{3}{4}}H^{\frac{5}{2}}T^{\frac{1}{4}}\Big). \\
&= O\Big(\sqrt{SAH^2T\iota} + H\sqrt{T\iota}\log(T) + \sqrt{SAH^2\beta T\iota} + SAH^3\sqrt{SN_0\iota}\log(T) \\
&\qquad + \sqrt{SAH^3\beta^2T\iota} + (SA\iota)^{\frac{3}{4}}H^{\frac{5}{2}}T^{\frac{1}{4}} + H^2SN_0\log(T)\Big) \\
&= O\Big(\sqrt{SAH^2T\iota} + H\sqrt{T\iota}\log(T) + SAH^3\sqrt{SN_0\iota}\log(T) + (SA\iota)^{\frac{3}{4}}H^{\frac{5}{2}}T^{\frac{1}{4}} + H^2SN_0\log(T)\Big) \\
&= O\Big(\sqrt{SAH^2T\iota} + H\sqrt{T\iota}\log(T) + S^2A^{\frac{3}{2}}H^6\iota\log(T) + (SA\iota)^{\frac{3}{4}}H^{\frac{5}{2}}T^{\frac{1}{4}} + S^2AH^8\iota\log(T)\Big) \\
&= O\Big(\sqrt{SAH^2T\iota} + H\sqrt{T\iota}\log(T) + S^2A^{\frac{3}{2}}H^8\iota T^{\frac{1}{4}}\Big).
\end{aligned}$$
(82)

## C  Other Results

### C.1  Local Switching Cost Analysis

The notion of local switching cost for RL is introduced in [Bai et al., 2019] to quantify the adaptivity of the learning algorithms. With a slight abuse of notations, we use $\pi_{k,h}$ to denote the policy at the $h$-th step of the $k$-th episode. We first recall formal definition of the local switching cost.

**Definition 1.** *The local switching cost at $(s, h)$ is defined as*

$$n_{\text{switch}}(s, h) := \sum_{k=1}^{K-1}\mathbb{I}\left[\pi_{k,h}(s) \neq \pi_{k+1,h}(s)\right].$$

*The total local switching cost is then defined as*

$$N_{\text{switch}} := \sum_{s\in\mathcal{S}}\sum_{h=1}^{H}n_{\text{switch}}(s, h).$$

Now we prove Theorem 2.

*Proof of Theorem 2.* By the definition of $e_i$, it is easy to verify that $e_{i+1} \geqslant (1 + \frac{1}{2H})e_i$ for any $i \geqslant 1$. Then the number of stages of $(s, a, h)$ is at most

$$\frac{\log(\frac{N_h^{K+1}(s,a)}{2H} + 1)}{\log(1 + \frac{1}{2H})} \leqslant 4H\log(\frac{N_h^{K+1}(s,a)}{2H} + 1).$$

Because $\pi_{k,h}(s) = \arg\max_a Q_h^k(s, a)$, we have that

$$\mathbb{I}\left[\pi_{k,h}(s) \neq \pi_{k+1,h}(s)\right] \implies \mathbb{I}\left[\exists a, Q_h^{k+1}(s, a) \neq Q_h^k(s, a)\right].$$

Now, by definition, we have that

$$n_{\text{switch}}(s, h) = \sum_{k=1}^{K-1} \mathbb{I}\left[\pi_{k,h}(s) \neq \pi_{k+1,h}(s)\right]$$

$$\leqslant \sum_{k=1}^{K-1} \mathbb{I}\left[\exists a, Q_h^{k+1}(s, a) \neq Q_h^k(s, a)\right]$$

$$\leqslant \sum_a 4H \log(\frac{N_h^{K+1}(s, a)}{2H} + 1).$$

Finally, by the concavity of $\log(x)$ in $x$, the total local switching cost of UCB-ADVANTAGE is bounded by

$$N_{\text{switch}} = \sum_{s \in \mathcal{S}} \sum_{h=1}^H n_{\text{switch}}(s, h)$$

$$\leqslant \sum_{s,a,h} 4H \log(\frac{N_h^{K+1}(s, a)}{2H} + 1)$$

$$\leqslant 4H^2 SA \log(\frac{T}{2SAH^2} + 1)$$

$$= O(H^2 SA \log(\frac{K}{SAH})).$$

$\square$

## C.2 Application to Concurrent RL

In concurrent RL, multiple agents act in parallel and shares the experience in a limited way to accelerate the learning process. In this subsection, we follow the setting in [Bai et al., 2019] to introduce the problem.

Suppose there are $M$ parallel agents, where each agent interacts with the environments independently. In the concurrent RL problem, each agent finishes an episode simultaneously, so that there are $M$ episodes done per concurrent round. The agents can only exchange experience and update their policies at the end of each round. The goal is to find an $\epsilon$-optimal policy using the minimum number of rounds, which we also refer to as the number of concurrent episodes.

In Algorithm 2, we present the details of the concurrent UCB-ADVANTAGE algorithm. The idea is to simulate the single-agent UCB-ADVANTAGE by treating the $M$ episodes finished in a single round as $M$ consecutive episodes (without policy change) in the single-agent setting. We collect the trajectories and feed them to the single-agent UCB-ADVANTAGE. When an update is triggered in the single-agent UCB-ADVANTAGE during an episode, we update the $Q$-function (as well as the value function) and discard the trajectories left in the round.

We now prove Corollary 3 that shows the performance of the concurrent UCB-ADVANTAGE.

*Proof of Corollary 3.* The proof follows the similar lines in the proof of Theorem 5 in [Bai et al., 2019]. By Theorem 2, the switching cost is at most $O(H^2 SA \log(\frac{K_\epsilon}{SAH}))$, so there are at most

$$O(H^2 SA \log(\frac{K_\epsilon}{SAH}) + \frac{K_\epsilon}{M}) = \tilde{O}(H^2 SA + \frac{H^3 SA}{\epsilon^2 M})$$

concurrent episodes. On the other hand, the regret incurred in the episodes corresponding to $K_\epsilon$ is at most $\tilde{O}(\sqrt{SAH^3 K_\epsilon}) \leqslant K_\epsilon \epsilon$, so by randomly choosing an episode index $k$ and selecting $\pi = \pi_k$ we achieve a policy with expected performance at most $\epsilon$ below the optimum. $\square$

## C.3 Lower Bound of the Sample Complexity

**Theorem 20.** *For any $H$, $S$, and $A$ greater than a universal constant, and all $\epsilon \in (0, \frac{8}{H}]$, for any algorithm with input parameter $\epsilon$, there exists an episodic MDP with $S$ states, $A$ actions, horizon $H$ such that, with probability at least $1/2$, among the execution history of the algorithm, there are at least $\Omega(SAH^3/\epsilon^2)$ episodes in which the corresponding policy $\pi_k$ satisfies that $V_1^*(s_1^k) - V_1^{\pi_k}(s_1^k) > \epsilon$.*

---

**Algorithm 2** Concurrent UCB-ADVANTAGE

---

**Initialize:** $Q_h(s,a) \leftarrow H-h+1$, $k \leftarrow 1$, $K_\epsilon \leftarrow \frac{c_5 SAH^3 \log(\frac{SAH}{\epsilon})}{\epsilon^2 M}$ ($c_5$ is a large enough universal constant).

**for** concurrent episodes $k = 1, 2, 3, \ldots$ **do**

    All agents follow the same policy $\pi_k$ where $\pi_{k,h}(s) = \arg\max_a Q_h(s,a)$.

    **for** $i = 1, 2, 3, \ldots, M$ **do**

        Collect the trajectory of the $i$-th agent and feed it to UCB-ADVANTAGE

        **if** an update is triggered **then**

            Update $Q$-value function following UCB-ADVANTAGE;

            **break**

        **end if**

    **end for**

    **if** The number of trajectories use is greater than or equal to $K_\epsilon$ **then**

        **break**

    **end if**

**end for**

---

*Proof Sketch.* Instead of presenting a concrete proof of Theorem 20, we provide the high-level intuition in the construction and analysis.

Like the regret lower bound analysis in [Jin et al., 2018], we consider the special case where $S = A = 2$. It does not require too much difficulty to generalize to arbitrary $S$ and $A$. Also, we will use almost the same hard instance as constructed in the proof of Theorem 3 in [Jin et al., 2018].

We recall the structure of "JAO MDP" in [Jaksch et al., 2010]. There are two states in the MDP, named $s_0$ and $s_1$. The rewards are defined as $r(s_0, a) = 0$ and $r(s_1, a) = 1$ for any $a$ and the transition probabilities are defined as $P(\cdot|s_1, a) = [\delta, 1-\delta]^\top, \forall a$, $P(\cdot|s_0, a) = [1-\delta, \delta]^\top, \forall a \neq a^*$ and $P(\cdot|s_0, a^*) = [1-\delta-\epsilon, \delta+\epsilon]^\top$. Clearly the optimal action for state $s_0$ is $a^*$. Let $\delta < \frac{1}{2}$ be fixed. By the lower bound of [Jaksch et al., 2010], there exists a constant $c_5 > 0$, such that for any $\epsilon \in (0, \frac{\delta}{2})$, it costs at least $c_5 \cdot \frac{\delta}{\epsilon^2}$ observations to identify $a^*$ with non-trivial probability.

By connecting $H$ JAO MDPs with different optimal actions layer by layer, we get an episodic MDP with horizon $H$. We choose $\delta = \frac{16}{H}$ to ensure that the MDP is well-mixed for $h \geqslant \frac{H}{2}$. For any $\epsilon \leqslant \frac{8}{H} = \frac{\delta}{2}$ and $h \geqslant \frac{H}{2}$, the agent reaches $s_0$ in the $h$-th layer with at least constant probability. If there are at least $\frac{7H}{8}$ layers in which the agent can not identify $a^*$, then the agent makes $\Omega(H)$ mistakes in the range $h \in [\frac{H}{2}, \frac{3H}{4}]$. Because each mistake for $h \in [\frac{H}{2}, \frac{3H}{4}]$ leads to $\Omega(\epsilon H)$ regret , the expected regret incurred during one episode is $\Omega(\epsilon H^2)$. As a result, if the total number of observations is less than $\frac{c_5 H}{8} \cdot \frac{\delta}{\epsilon^2}$ (i.e., number of episodes less than $\frac{c_5}{8} \cdot \frac{\delta}{\epsilon^2}$), the expected regret per episode is $\Omega(\epsilon H^2)$. Replacing $\epsilon$ by $\epsilon H^2$, we have that for the first $\Theta(\delta H^4/\epsilon^2) = \Theta(H^3/\epsilon^2)$ episodes, the expected regret per episode is $\Omega(\epsilon)$. The proof is then completed by applying Markov's inequality.

$\square$