[Reviews · NeurIPS 2020]

Review 1

Summary and Contributions: This paper proposes UCB-Advantage, a model-free RL algorithm, which achieves nearly optimal regret upper bound in the episodic MDP setting. =====Update===== Thanks for the rebuttal. I think this paper provides useful techniques which could be promising in studying more complicated settings. I stick to my current rating.

Strengths: The main contribution of this paper is to reduce a \sqrt{H} factor in the upper bound of Q-learning with UCB in Jin et al. 2018, matching the best model-based RL algorithm UCBVI and closing the gap between the lower bound up to \log{T} factor. The idea is to combine the learning rate scheme of Jin et al., 2018 and a reference-advantage update rule. Then the UCB is carefully designed to estimate the reference and advantage terms. The authors also discussed the lowing switching cost and concurrent RL settings. The technique used, especially the reference-advantage update (with the stage-based update and finally they worked together) is impressive and novel.

Weaknesses: Can the proposed techniques, especially this reference-advantage update benefit the recent popular study of RL and planning with features, e.g., linear models, linear value functions, and block / low rank MDPs?

Correctness: The ideas and techniques look reasonable and correct to me (although I did not check every detail of the proofs).

Clarity: This paper is well written. The authors tried to make the presentation and explanation clear.

Relation to Prior Work: Mostly the related work is well discussed. The following paper is related and missing when mentioning the best model-based algorithms: Policy Certificates: Towards Accountable Reinforcement Learning, Dann et al., 2019.

Reproducibility: Yes

Additional Feedback:


Review 2

Summary and Contributions: This paper proposes a novel algorithm "UCB-ADVANTAGE" in the finite horizon tabular MDP setting. This paper shows that the regret bound of the proposed algorithm is tight and almost matches the lower bound. Moreover, the theoretical result in this paper indicate that model-free algorithm can has almost the same regret bound as the model-based algorithms Azar et al. 2017, Kakade et al. 2018, and such an observation is very important to the community.

Strengths: The major novelty of this paper is in introducing the reference-advantage decomposition technique to help improve the regret bound of UCB-Bernstein algorithm in Jin et al. (2018). With such a technique, it is the first time that the model-free algorithm can be shown to perform as good as model-based algorithms. Moreover, it seems that the reference-advantage decomposition technique is potentially applicable to other RL problems to improve the state-of-art regret bounds in the tabular setting. I think the contribution of this paper is significant.

Weaknesses: Considering the specific setting this paper focus, the theoretical result is solid and there is no obvious weakness and limitation of this work. Just one minor point: In algorithm 1, what is the dependence of the "large enough constants" c_1, c_2, c_3, c_4 on the problem setting? Could the author provide some guidance of choosing these parameters in practice (for example, some lower bounds of these constants)?

Correctness: The claims and method are correct.

Clarity: Overall the paper is well written and easy to follow. One suggestion is that it might be easier for readers to understand the main result if the author can specify the parameter setting directly in the main theorem.

Relation to Prior Work: This paper gives a detailed comparsion with previous works of both model free and model based RL. The contribution of this work is clear.

Reproducibility: Yes

Additional Feedback: The authors’ response has addressed my questions. I think the contribution of this paper is significant, thus I will improve my evaluation to accept.


Review 3

Summary and Contributions: The paper develops regret bounds for finite state/action, model free (MF) reinforcement learning (RL), in an episodic setting based on online on-policy updates. At the beginning of each episode the learner observes the initial state that may be adversarial. A novel algorithm termed UCB-advantage is developed that achieves the best regret obtained up to now, and achieves the information-theoretic lower bound (up to logarithmic factors). The bound improves on a previous 2018 bound by Jin et el., by removing an extra factor of \sqrt{H}, where H is the number of steps per episode. The paper answers an open question as to whether a model free algorithm can achieve the same efficiency as model based (MB) approaches, which require higher computational efforts. In fact, this is the first \sqrt{T} regret rate demonstrated for model free RL.

Strengths: The paper solves an open problem in RL, namely whether MF learning can be as sample efficient as MB learning. This is in itself a significant achievement, especially as the bounds are in probability rather than in expectation. The algorithm proposed is novel, although belonging to the family of UCB algorithms (namely, incorporating an exploration bonus). It differs from the latter by introducing a new notion of reference-decomposition that breaks up the value into a reference component and an advantage component (that, somewhat confusingly, differs from the standard notion of advantage in RL). Furthermore, the authors use a novel stage-based approach that allows them to integrate the standard Q update with the one resulting from the advantage function. This allows to ensure a low frequency of policy switches during execution, and to remove the \sqrt{H} factor in the bound. Finally, the authors demonstrate the applicability of their results to concurrent RL.

Weaknesses: Given the complex notations and multiple definitions, it would be helpful to include a table in the appendix summarizing the notation.

Correctness: I have not reviewed the detailed proofs in the appendix, but did go through the proof of Theorem 1 presented in the main text. The proof is broken down into several logical steps, most of which are proved in the appendix. The initial part of the proof demonstrates that the reference value at time k provides a good approximation to the optimal value function (Corollary 6 proved in the appendix with other relevant lemmas). The proof of Theorem 1, conditioned on the successful events established in the previous lemmas, makes use of martingale concentration inequalities, that have become prevalent in this kind of analysis.

Clarity: The paper is highly technical yet very clearly written. The authors have taken great care to motivate their work, to provide the requisite background and to outline the intuition of the algorithm and the outline of the proofs, most of which have been relegated to the appendix for obvious reasons.

Relation to Prior Work: The closest work to the present is by Jin et al. from 2018 and is amply referred to, as are other relevant works. A related work alluded to by Dong et al. 2019 deals with PAC, rather than, regret bounds. As noted by the authors, the topic of regret bounds for MF RL has received less attention that that of MB RL.

Reproducibility: Yes

Additional Feedback: Comments following rebuttal: I have read the rebuttal and maintain my previous assessment of the paper.

[Author Response · NeurIPS 2020]

Many thanks for your valuable comments and suggestions. We will address the minor comments (such as to fix the typos, add the discussion about the parameters, and broaden the related works section) in the next version of the paper. We now respond to the major comments are as follows.

**To Reviewer 2:** We believe that the proposed techniques could be helpful in other RL settings such as continuous state spaces. Take RL with the linear model as an example. In this case, to learn $Q^*$, it suffices to learn the $d$-dimensional vector $w^*$ (which we name as pseudo value function) because $Q_h^*(s, a) = \phi(s, a)^\top w_h^*$, where $\phi(s, a)$ is the $d$-dimensional feature vector of $(s, a)$. To reduce the estimation variance via reference-advantage decomposition, we first aim at learning a rough estimation of $w^*$ as the reference pseudo value function, using time steps that are only polynomially depending on $A$, $H$, $d$, $1/\epsilon$, $\ln(1/p)$, $\ln T$, where $p$ is the failure probability, and $\epsilon$ is the accuracy parameter. More formally, we believe that the key is to prove an analogue of Lemma 5 for the linear model. We will add this discussion in the next version of the paper.

We will also discuss the work on policy certificates (Dann et al., 2019) in related work section.

**To Reviewer 3:** At line 397, we mentioned that $[c_1, c_2, c_3] = [2, 2, 5]$ is sufficient. The safe choice in the current version of the paper for $c_4$ is $c_4 = 57603$, which seems quite large because we bound $(1 + \frac{1}{H})$ by 2 in many places for convenience. When $H$ is sufficiently large (say greater or equal to 200), we can bring the choice of $c_4$ to around 200. We will add this discussion ot the next version of the paper.

**To Reviewer 4:** Thank you for your suggestion. We will include the table for all notations in the next version of the paper.

[Meta-Review · NeurIPS 2020]

The paper shows a model-free algorithm with an improved regret bound for finite-state finite-horizon MDP problems. The new bound closes the gap with the best model-based result. This is a nice theoretical contribution.